# Impaired neuronal sodium channels cause intranodal conduction failure and reentrant arrhythmias in human sinoatrial node

Ning Li [1,2,8], Anuradha Kalyanasundaram[1,2,8], Brian J. Hansen[1,2,8], Esthela J. Artiga[1,2], Roshan Sharma[3], Suhaib H. Abudulwahed[1,2], Katelynn M. Helfrich[1,2], Galina Rozenberg[1,2], Pei-Jung Wu[1,2], Stanislav Zakharkin[1], Sandor Gyorke[1,2], Paul ML. Janssen[1,2], Bryan A. Whitson[2,4], Nahush A. Mokadam[2,4], Brandon J. Biesiadecki[1,2], Federica Accornero[1,2], John D. Hummel[2,5], Peter J. Mohler[1,2], Halina Dobrzynski [6,7], Jichao Zhao [3] & Vadim V. Fedorov [1,2]*

Mechanisms for human sinoatrial node (SAN) dysfunction are poorly understood and whether human SAN excitability requires voltage-gated sodium channels (Nav) remains controversial. Here, we report that neuronal (n)Nav blockade and selective nNav1.6 blockade during high-resolution optical mapping in explanted human hearts depress intranodal SAN conduction, which worsens during autonomic stimulation and overdrive suppression to conduction failure. Partial cardiac (c)Nav blockade further impairs automaticity and intra-nodal conduction, leading to beat-to-beat variability and reentry. Multiple nNav transcripts are higher in SAN vs atria; heterogeneous alterations of several isoforms, specifically nNav1.6, are associated with heart failure and chronic alcohol consumption. In silico simulations of Nav distributions suggest that $I_{Na}$ is essential for SAN conduction, especially in fibrotic failing hearts. Our results reveal that not only cNav but nNav are also integral for preventing disease-induced failure in human SAN intranodal conduction. Disease-impaired nNav may underlie patient-specific SAN dysfunctions and should be considered to treat arrhythmias.

---

[1] Department of Physiology and Cell Biology, The Ohio State University Wexner Medical Center, Columbus, OH, USA. [2] Bob and Corrine Frick Center for Heart Failure and Arrhythmia, Davis Heart and Lung Research Institute, The Ohio State University Wexner Medical Center, Columbus, OH, USA. [3] Auckland Bioengineering Institute, The University of Auckland, Auckland, New Zealand. [4] Department of Surgery, The Ohio State University Wexner Medical Center, Columbus, OH, USA. [5] Department of Internal Medicine, The Ohio State University Wexner Medical Center, Columbus, OH, USA. [6] Division of Cardiovascular Sciences, The University of Manchester, Manchester, UK. [7] Department of Anatomy, Jagiellonian University Medical College, Cracow, Poland. [8] These authors contributed equally: Ning Li, Anuradha Kalyanasundaram, Brian J. Hansen *email: vadim.fedorov@osumc.edu

Normal cardiac rhythm is maintained by the human sinoatrial node (SAN) complex, which is compartmentalized into multiple intranodal pacemakers and conduction pathways (sinoatrial conduction pathways, SACPs) within an intramural three-dimensional (3D) structure[1–3]. Robust SAN pacemaking and conduction is maintained amidst a plethora of internal and external perturbations by several ion channels, which are heterogeneously expressed within the SAN compartments[3–5]. Impairment of pacemaking or conduction between SAN compartments can result from a variety of cardiac pathologies or extrinsic factors (e.g., antiarrhythmic drugs) and may lead to SAN dysfunction (SND), a disease that can significantly impact quality of life[6]. In addition to rhythm abnormalities, SND can often facilitate deterioration to atrial fibrillation (AF)[7] and heart failure (HF)[8].

Currently, patients with symptomatic SND rely on a single treatment option of electrical pacemaker implantation[9], which acts as a crutch to support but not heal the heart. The development of optimal alternative treatments for SND will require in-depth knowledge of the mechanisms involved in robust human SAN rhythm regulation. However, there is a paucity of studies addressing mechanisms that contribute to automaticity and intranodal conduction directly in the human SAN complex at the molecular, cellular, and tissue levels. Disease-induced remodeling of many of the molecular components critical to SAN function including hyperpolarization-activated cyclic nucleotide-gated (HCN) channels and G-protein-coupled inwardly rectifying potassium channels, adenosine receptors (A1R), and L-type calcium channels, as well as structural fibrotic remodeling can lead to SND[3,10–13]. However, majority of these molecular components critical to SAN function have been studied only in animal models[5], which have significantly different functional and anatomical features compared with the human SAN, especially when studying aged/diseased human SAN with SND. In particular, the roles of voltage-gated sodium channels (Nav), which are major contributors to cardiac and neuronal excitability, remain controversial in human SAN pacemaking and conduction. Although the presence of Nav has been reported in the SAN from multiple species[14–16], direct evidence confirming the expression and functional role of cardiac (cNav) and/or neuronal (nNav) isoforms in the human SAN is lacking[17]. Some patients with symptomatic SND are found to harbor loss-of-function mutations in the SCN5A gene, which encodes the α-subunit of the cardiac isoform Nav1.5 (ref.[18]), suggesting that functional voltage-gated Na$^+$ current ($I_{Na}$) may be necessary for maintaining human SAN pacemaking and conduction. Furthermore, although antiarrhythmic drugs that block Nav have been shown to inadvertently unmask SND in patients predisposed to the arrhythmia[19–21], no study has determined the specific Nav isoform(s) mediating this detrimental effect on the human SAN. Hence, there is a critical need to determine the functional contributions of Nav channels in maintaining SAN robustness, by studying their role directly in the unique 3D human SAN complex[2–4].

Therefore, the primary goal of the current study is to determine the existence and specific role of nNav and cNav isoforms in human SAN pacemaking and intranodal conduction, and to reveal their disease-induced alterations in explanted human hearts. Some of the comorbidities associated with the explanted human hearts used in this study include, but are not limited to, HF, AF, hypertension (HTN), and modifying risk factors such as smoking, chronic alcohol consumption, and drug abuse (Supplementary Tables 1 and 2). We employed high-resolution near-infrared sub-surface optical mapping, the only approach currently able to reveal intramural human SAN pacemaking and conduction. Furthermore, our functional studies are complemented by human SAN computational simulations and molecular mapping of multiple nNav and cNav transcripts and protein expression.

Here, we report that in the human SAN, nNav mainly contribute to intranodal conduction, whereas cNav play dual roles in both pacemaking and conduction. Impairment of Nav can lead not only to depression of SAN pacemaker and conduction but also to a perfect storm of SAN exit block, disorganized intranodal pacemakers, and SAN micro- and macro-reentry. Our data also suggest that by altering nNav in the SAN and/or atria, HF and chronic alcohol consumption could promote a patient-specific, mechanistic substrate for tachy-brady arrhythmias and SAN conduction blocks.

## Results

**Essential roles of Navs under physiological conditions**. Under control conditions, all optically mapped human hearts (non-failing $n = 12$ and HF $n = 2$) exhibited stable intrinsic sinus rhythm of 56 to 116 beats per minute, with a sinus cycle length (SCL) of $729 \pm 200$ ms (Supplementary Table 1 and Supplementary Fig. 1). These values are comparable with "intrinsic" rates reported in vivo during autonomic blockade in adult human subjects with/without cardiac comorbidities[22–24]. For more details on the stability and experimental protocols, please see the Methods section. All functional data presented below are averages of all human SAN preparations (HF and non-failing) studied for each specific drug protocol. As the small sample size precludes subset analysis, heart-specific data are presented in Supplementary Fig. 1.

Tetrodotoxin (TTX) dose-dependently prolonged SCL by $6 \pm 5\%$ (mean $\pm$ SD, $n = 8$, $P < 0.05$, Wilcoxon's test) at 100 nM and $43 \pm 41\%$ ($n = 7$, $P < 0.05$, Wilcoxon's test) at 1–3 μM, and caused shifts of the intranodal leading pacemaker and preferential SACP (Fig.1; Table 1). Sinoatrial conduction time at sinus rhythm (SACTsr) was increased to ~250% ($n = 6$, $P < 0.05$, Wilcoxon's test) by TTX 100 nM and to ~610% ($n = 6$, $P < 0.05$, $t$-test) by TTX 1–3 μM. Figure 2a, b shows the effects of TTX 1 μM on SCL and SACTsr. Furthermore, TTX 1–3 μM significantly increased SACTsr/SCL ratio from 8% to 23% ($P < 0.01$), suggesting that prolongation of atrial CL by cNav blockade was primarily due to depression of SAN conduction rather than automaticity.

Nav blockade also prolonged intranodal conduction (measured from the SAN leading pacemaker to the SAN border) by 31% and 47% ($P < 0.05$, $t$-test) at TTX 100 nM and 1–3 μM, respectively (Fig. 2c, d). Decremental SAN conduction (Fig. 2c) gradually increased SACTsr and eventually caused exit block (5/7 hearts) and complete atrial arrest (3/7 hearts) at TTX 1–3 μM. We also observed significantly higher beat-to-beat variations of SCL and SACT during TTX 1–3 μM than at control conditions or TTX 100 nM (Fig. 2e and Supplementary Fig. 2).

To eliminate the possibility of natural drift of tissue properties affecting our results during the experimental mapping period of 2 h, we verified in all studied human SAN experiments as well as in previous experiments[3] that the drift at baseline conditions in SCL was ~5% per hour, whereas drift was not observed in atrial conduction velocity (CV), which suggests that changes in automaticity and conduction seen with low- and high-dose TTX, respectively, were significant and not due to time-dependent changes in tissue properties (Supplementary Fig. 3).

**Protective roles of Navs during stress in the human SAN**. We next investigated the contribution of Nav in protecting SAN conduction during metabolic challenge with adenosine bolus or overdrive suppression by fast atrial pacing, which are clinically employed tests to unmask SND[25,26]. Adenosine bolus was

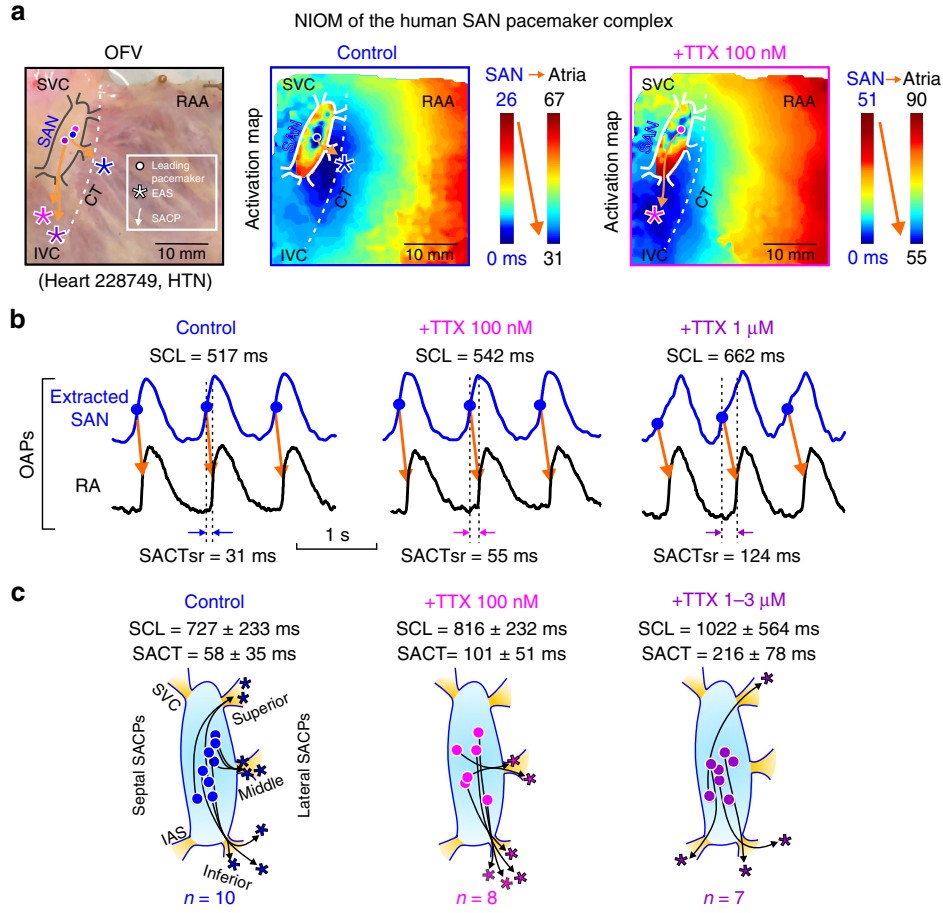

**Fig. 1 High-resolution near-infrared optical mapping (NIOM) reveals effect of nNav and cNav blockade on human SAN intramural activation. a** Left: human SAN preparation and optical mapping field of view (OFV). Activation maps of Heart 228749 at control (middle) and during 100 nM TTX perfusion (right). Dot indicates the leading pacemaker inside the sinoatrial node (SAN); asterisk indicates the earliest atrial activation site (EAS); arrow indicates the sinoatrial conduction pathway (SACP). **b** TTX effects on sinus cycle length (SCL) and sinoatrial conduction time at sinus rhythm (SACTsr). **c** Locations of SAN leading pacemaker (dots), earliest atrial activation site, and sinoatrial conduction pathways at control (left, $n = 10$), during perfusion of 100 nM (middle, $n = 8$), and 1–3 µM TTX (right, $n = 7$). Data presented as mean ± SD. CT crista terminalis, n/cNav neuronal/cardiac sodium channels, OAP optical action potential, RA (A) right atrial (appendage), S/IVC superior/inferior vena cava, TTX tetrodotoxin. Source data are provided as a Source Data file.

**Table 1 Effects of Nav blockade on human SAN and atrial conduction**

|  | 100 nM TTX % Control ± SD | n | P-value | 1–3 µM TTX % Control ± SD | n | P-value | Nav1.6 blocker % Control ± SD | n | P-value |
|---|---|---|---|---|---|---|---|---|---|
| SCL | 106 ± 5% | 8 | 0.0141 | 143 ± 41% | 7 | 0.0156 | 106 ± 6% | 5 | 0.1000 |
| SACTsr | 250 ± 201% | 6 | 0.0313 | 610 ± 355% | 6 | 0.017 | 120% | 1 |  |
| 2 Hz SACTppb | 319 ± 148% | 6 | 0.0152 | 407 ± 272% | 5 | 0.0651 | 1277% | 1 |  |
| 2 Hz cSNRTi | 236 ± 234% | 7 | 0.1763 | 139 ± 118% | 3 | 0.6224 | 226 ± 259% | 5 | 0.3125 |
| 2 Hz cSNRTd | 137 ± 185% | 6 | 0.8438 | 4 ± 141% | 4 | 0.2651 | 17% | 1 |  |
| RA CV longitudinal | 94 ± 4% | 5 | 0.0327 | 70 ± 14% | 6 | 0.0036 | 94 ± 10% | 5 | 0.2684 |
| RA CV transverse | 88 ± 19% | 5 | 0.0074 | 68 ± 23% | 6 | 0.0212 | 100 ± 10% | 5 | 0.9590 |

*cSNRTd/i corrected direct/indirect sinus node recovery time, RA right atrial, SACTsr/ppb sinoatrial conduction time at sinus rhythm/post-pacing beat, SAN sinoatrial node, SCL sinus cycle length. Percent data are reported as mean ± SD. Normality assumption was verified using Shapiro–Wilk test. Parametric data were analyzed with two-sided t-test. Non-parametric data were analyzed with two-sided Wilcoxon's test. Source data are provided as a Source Data file*

injected via the coronary artery in eight SAN preparations, which caused temporary hyperpolarization of the cardiomyocytes. At control conditions, adenosine bolus increased the maximum SCL by 30% ± 20%, whereas after 100 nM TTX, the same dose of adenosine prolonged SCL by 72 ± 54% ($n = 8$, $P < 0.05$, t-test) (Fig. 3). Moreover, SAN exit block was observed in five of eight hearts during adenosine bolus at 100 nM TTX, which did not occur at control conditions (Fig. 3).

Based on our molecular data that showed transcripts of nNav1.6 isoform to be higher in the SAN compared with other nNav isoforms (see below), we studied its unique role in SAN pacemaking and conduction using a specific nNav1.6 blocker 30 nM 4,9-Anhydrotetrodotoxin[27] in five human SAN experiments (Fig. 3c and Supplementary Fig. 4). These experiments revealed that selective blockade of nNav1.6 channels did not affect SCL but could depress SAN conduction and induced SAN exit block in

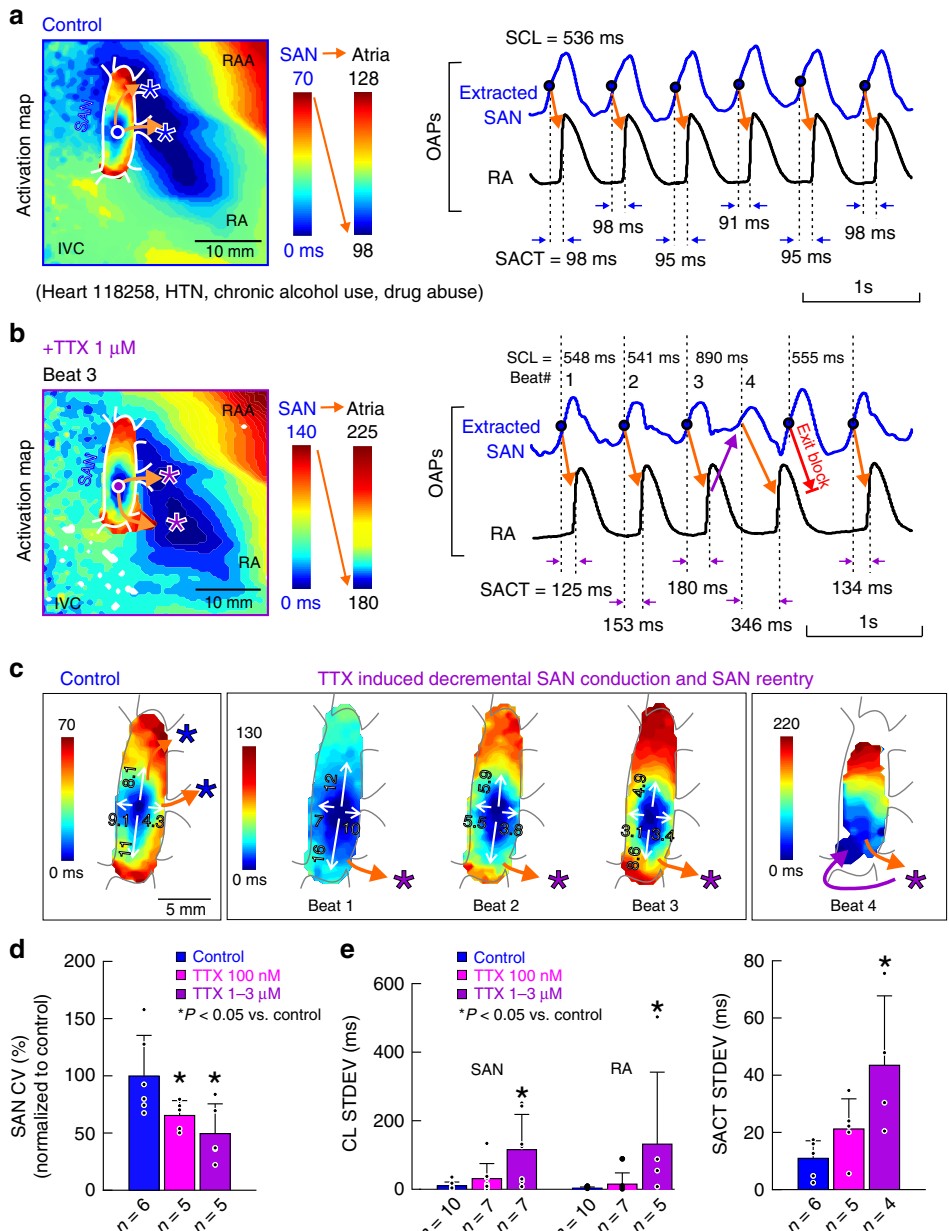

**Fig. 2 SAN intranodal conduction impaired by nNav and cNav blockade.** Activation map (left) and OAP recordings (right) show sinoatrial node (SAN) activation and conduction at control (**a**), and after blocking cardiac Nav with 1 μM TTX (**b**). Dot indicates the leading pacemaker inside the SAN; asterisk indicates the earliest atrial activation site; arrow indicates the SAN conduction pathway. **c** Left: SAN activation map showing intranodal conduction at control condition. Middle to right: SAN activation maps of four sequential beats showing the gradual decrease of intranodal conduction and SAN macro-reentry after cardiac Nav blockade with 1 μM TTX. White arrows indicate the direction of intranodal conduction; numbers beside arrows denote SAN conduction velocity (cm/s). **d** Summary of SAN conduction velocity (CV) changes at 100 nM and 1–3 μM TTX (n = 5) compared with control (n = 6). **e** TTX increased the beat-to-beat variation shown as SD (STDEV) of SAN (control: n = 10; TTX 100 nM and 1–3 μM: n = 7) and atrial (control: n = 10; TTX 100 nM: n = 7; 1–3 μM: n = 5) cycle length (CL), and sinoatrial conduction time (SACT) during control (n = 6), TTX 100 nM (n = 5), and TTX 1–3 μM (n = 4). Data presented as mean ± SD. IVC inferior vena cava, OAP optical action potential, RA (A) right atrial (appendage), TTX tetrodotoxin. Data in **d** and **e** were represented in mean ± SD. Asterisks in **d** and **e** indicate P < 0.05 vs. control by two-tailed t-test. Source data are provided as a Source Data file.

three out of five experiments, during adenosine bolus and overdrive suppression; these results were comparable to the effects of 100 nM TTX (Fig. 3c). Our findings suggest that nNav1.6 blockade could be one of the contributors of the 100 nM TTX inhibitory effect on SAN conduction, thereby emphasizing its role as an important functional nNav isoform for human SAN conduction. We also evaluated the effect of TTX and nNav1.6 blocker on atrial conduction during pacing at 500 ms; nNav1.6 did not have a significant effect on both longitudinal and

transversal CV but TTX depressed longitudinal atrial CV by 6% and 30% (P < 0.01, t-test) at 100 nM and 1–3 μM concentrations, respectively (Table 1 and Supplementary Fig. 5).

At control conditions, atrial pacing at 500 ms did not unmask SND (cSNRTi > 525 ms) in any of the hearts tested (n = 14). After selectively blocking nNav with TTX 100 nM, the same rate of atrial pacing increased cSNRTi by 236 ± 234% (n = 7, P = 0.176, Wilcoxon's test) and unmasked SND in two hearts (HF 421856 and non-failing 642519; Supplementary Table 1) with

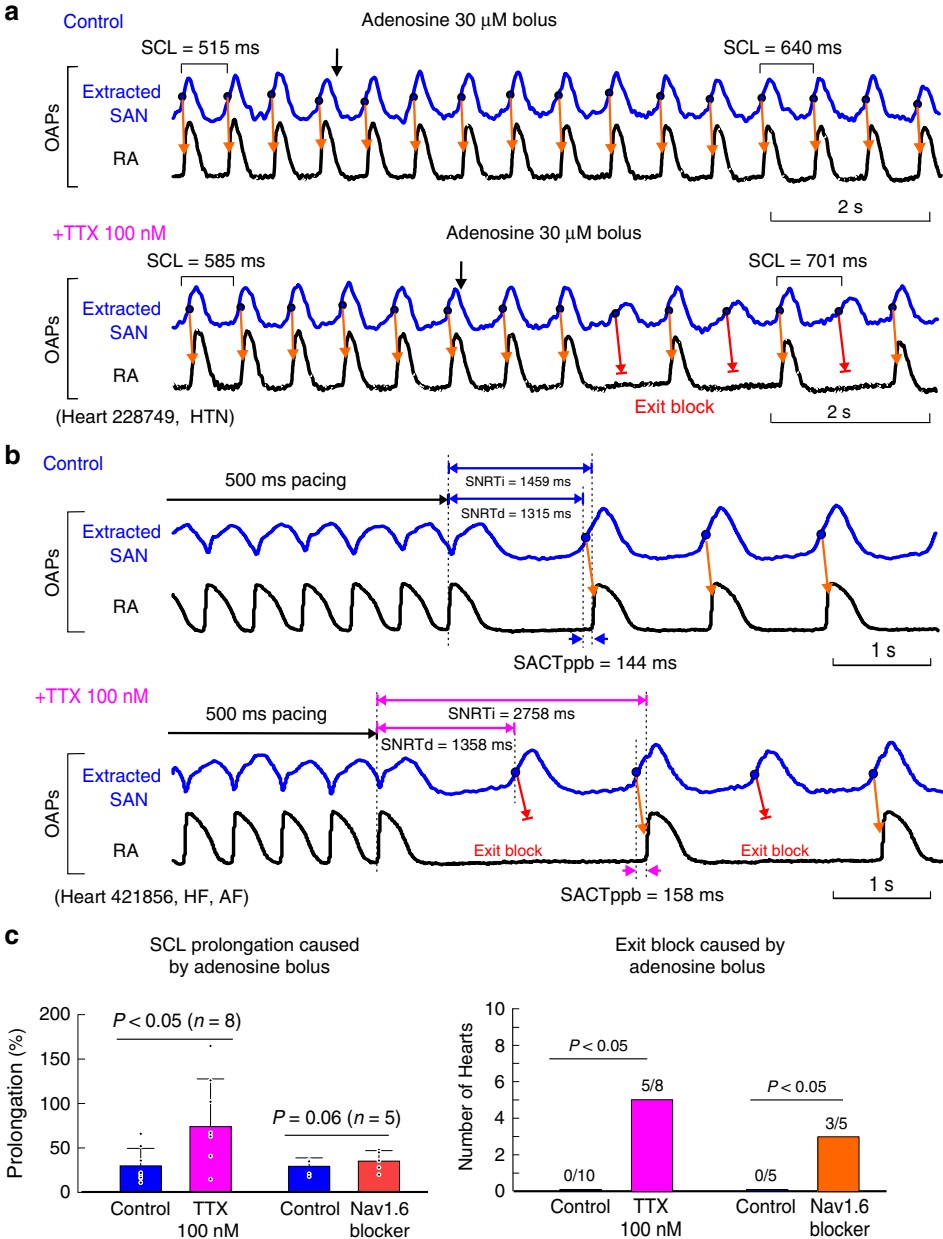

**Fig. 3 Protective role of nNav in human SAN conduction during hyperpolarization and overdrive pacing. a** Top: optical action potentials (OAPs) show a slight increase of sinus cycle length (SCL) during SAN hyperpolarization with 30 μM adenosine bolus at control conditions. Bottom: OAPs show exit block from the SAN to the RA after neuronal Nav blockade with 100 nM TTX during adenosine bolus. **b** Top: OAPs show normal SNRT and 1:1 conduction from the SAN to the RA after 500 ms atrial pacing at control conditions; bottom: OAPs show prolonged SNRT and 2:1 exit block from the SAN to the RA after 500 ms atrial pacing during 100 nM TTX pefusion. **c** Left: comparison of SCL prolongation caused by adenosine bolus at control conditions (*n* = 8), 30 nM nNav1.6 blocker (*n* = 5), and during 100 nM TTX perfusion (*n* = 8); right: exit block events observed at control conditions, 30 nM nNav1.6 blocker and during 100 nM TTX perfusion after adenosine bolus. ppb post-pacing beat, RA right atria, SAN sinoatrial node, SNRTi/d indirect/direct sinoatrial recovery time, TTX tetrodotoxin. Data of SCL prolongation in **c** were represented in mean ± SD, *n* = 8, *\*P* < 0.05, by *t*-test. Analysis of exit block events used two-sided proportion test with Yates continuity correction. Source data are provided as a Source Data file.

cSNRTi > 1400 ms due to exit block (Fig. 3; Table 1). Faster, 300 ms atrial pacing caused exit block in 3/15 experiments at control conditions, in 2/5 experiments with nNav1.6 blocker, in 3/7 experiments at TTX 100 nM, and in 5/6 hearts at TTX 1–3 μM. Figure 4 shows an example of conduction failure/arrhythmias, resulting from a perfect storm of stochastic SAN exit block, competing intranodal pacemakers, SAN micro- and macro-reentry, seen in two hearts (non-failing 118258 with history of chronic HTN/chronic alcohol consumption and HF 930597; Supplementary Table 1). In these hearts, preferential slowing of

SAN conduction by Nav blockade, led to intranodal unidirectional blocks and initiated intranodal micro-reentry or macro-reentry (Supplementary Movies 1 and 2). The micro-reentry pivot waves anchored to the longitudinal block region can produce both tachycardia and paradoxical bradycardia (due to exit block), despite an atrial activation pattern and ECG morphology identical to regular sinus rhythm[28]. Intranodal longitudinal conduction blocks usually coincided with interstitial fibrosis strands and SAN artery as in the case of heart 118258 as shown by histology in Fig. 4c. SAN reentrant arrhythmias were not

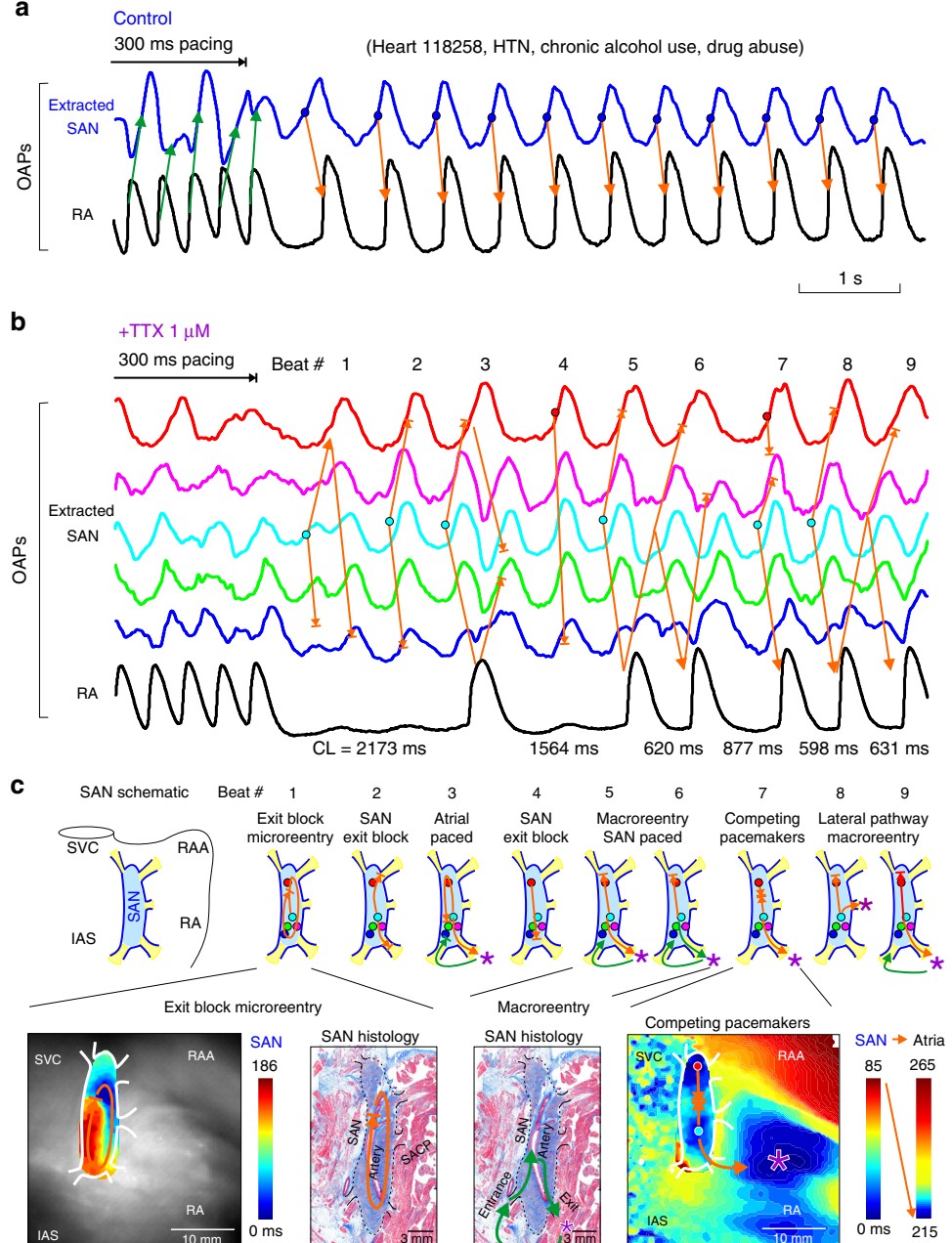

**Fig. 4 Pacing-induced SAN conduction failure and reentrant arrhythmias during nNav and cNav blockade. a** Optical action potentials (OAPs) show regular sinus rhythm and sinoatrial conduciton after 3.3 Hz atrial pacing at control. **b** OAPs show irregular sinus rhythm, atrial pause, and arrhythmia after 3.3 Hz atrial pacing during 1 μM TTX perfusion. **c** SAN activation and conduction pattern revealed by high-resolution optical mapping could explain the mechanism underlying the post-pacing reentrant arrhythmia during cardiac Nav blockade by 1 μM TTX. Top, schematic of the SAN pacemaker complex representing activation pattern for 9 post-pacing beats. Bottom, SAN activation maps and corresponding histology sections show locations of micro-reentry and macro-reentrant pathways within the SAN complex stucture. Colored dots indicate positions of sinoatrial (SAN) OAPs presented in **b**. Orange arrows show the conduction direction within the SAN and green arrows indicate conduction outside the SAN; asterisk indicates the earliest atrial activation site. IAS interatrial septum, RA(A) right atrial (appendage), S/IVC superior/inferior vena cava, SAN sinoatrial node, TTX tetrodotoxin.

observed at control conditions or with TTX 100 nM in any hearts, except in one (957855) experiment during nNa1.6 blockade.

**mRNA and protein profiles of Nav in human SAN and atria.** We next studied the relative mRNA and protein expression levels of Nav in human SAN tissues, compared with neighboring atria (Supplementary Table 3). The distribution of Nav isoforms, including nine α-subunits, and four β-subunits, were analyzed in

human SAN and surrounding atrial tissues by quantitative PCR (qPCR). With the exception of *SCN10A* (Nav1.8), which was detected only at low levels in some of the human RA (see Source Data file), all other isoforms were detected in human SAN as well as in RA at the mRNA level (Fig. 5). *SCN5A* (Nav1.5) and *SCN1B* (Navβ1) were the most abundantly transcribed Nav channel α- and β-subunits in both human SAN and RA, with higher levels in RA than in SAN. In contrast, mRNA transcripts of neuronal subunits *SCN1A* (Nav1.1), *SCN2A* (Nav1.2), *SCN8A* (Nav1.6),

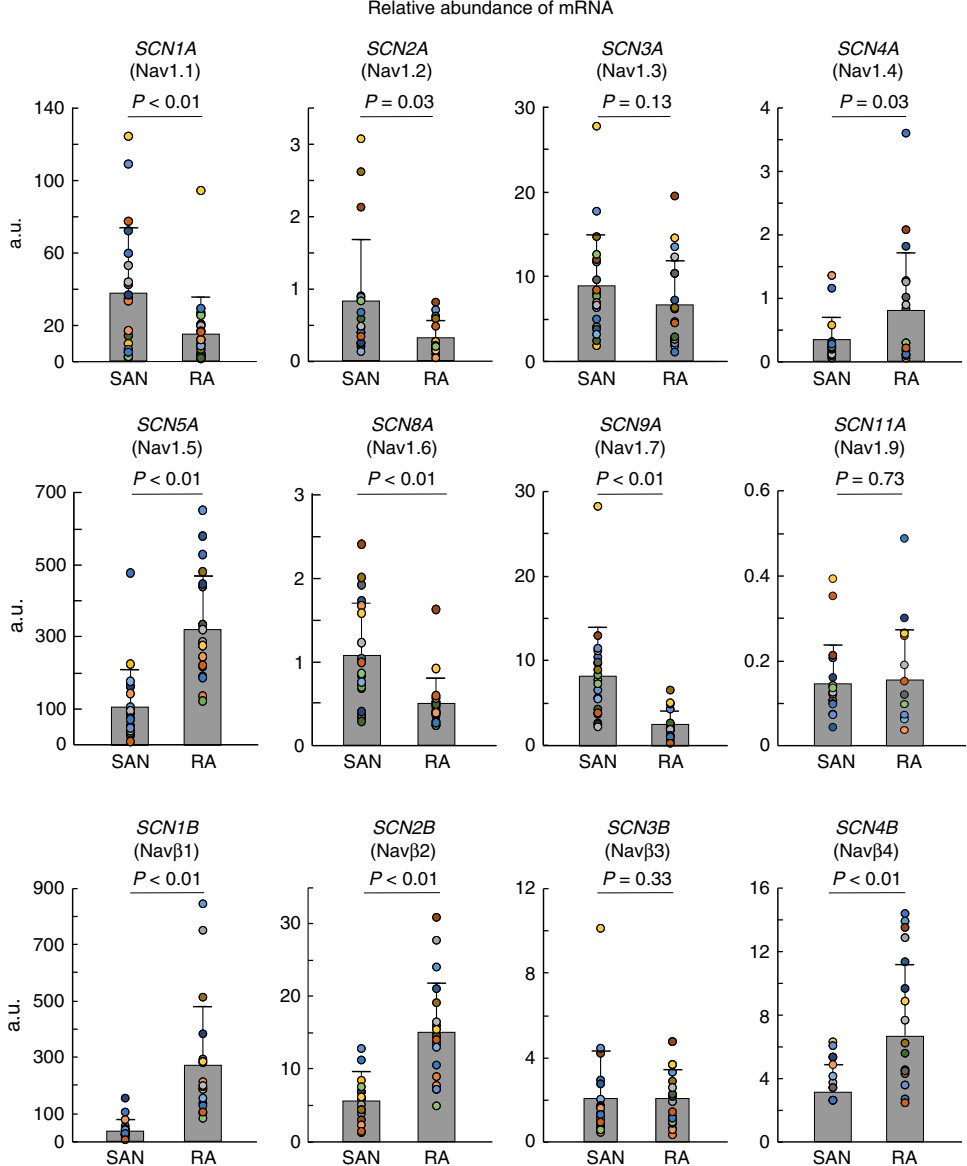

Relative abundance of mRNA

**Fig. 5 Distribution of cNav and nNav subunit mRNA in the human SAN and RA.** Summary data of qPCR analysis of different Nav isoform mRNA levels in the human sinoatrial node (SAN) and right atrial (RA) tissue; colored circles indicate individual data points for each human heart. All mRNA values were normalized to the ribosomal 18s gene. Nav voltage-gated sodium channels. Data were represented in mean ± SD; $n = 20$ for each group; normality was tested with Shapiro–Wilk test; two-sided $t$-test was used for parametric data and Wilcoxon's test was used for non-parametric data. Source data are provided as a Source Data file.

and SCN9A (Nav1.7) were higher in the human SAN than RA. In addition, we also quantified the transcript levels of other major proteins involved in human SAN pacemaking and conduction including HCN1, HCN4, GJA1 (Cx43), GJA5 (Cx40), and calcium channels CACNA1C (Cav1.2), CACNA1D (Cav1.3), and CAC-NA1G (Cav3.1). CACNA1D transcripts were significantly higher in the SAN vs. RA; as expected, we found higher levels of both HCN1 and HCN4, and lower levels of GJA1 transcripts in SAN vs. atria, which validated the purity of SAN tissue used for these studies (Supplementary Table 3).

Protein distribution patterns of cNav1.5 and nNav1.6 in human SAN and RA were also detected by immunostaining (Fig. 6a, b). Fluorescence density analysis confirmed that protein levels of cNav1.5 channels are higher in RA than in SAN cardiomyocytes ($P < 0.01$, pairwise test), whereas the nNav1.6 protein expression is higher in the SAN than RA ($P < 0.01$,

pairwise test). Localization of cNav1.5 and nNav1.6 is shown to be cardiomyocyte specific (Fig. 6c, d and Supplementary Fig. 6). Figure 6e shows that the cNav1.5 protein expression pattern detected by western blotting is consistent with both mRNA expression and immunostaining results. These results collectively confirm the presence of both nNav and cNav isoforms in the human SAN at the molecular level.

In addition, to correlate functional results with biochemical findings the expression levels of cNav1.5 and nNav1.6 were quantified in immunostained cryo-frozen tissue sections from three hearts functionally mapped with the nNav1.6 blocker (957855, 283273, and 670263) (Supplementary Fig. 7). Our results show that in heart 957855, which developed exit blocks with nNav1.6 blocker (Supplementary Fig. 6), the ratio of nNav1.6 between SAN and RA was the highest compared with two other hearts, which exhibited less sensitivity to nNav1.6 blocker. These

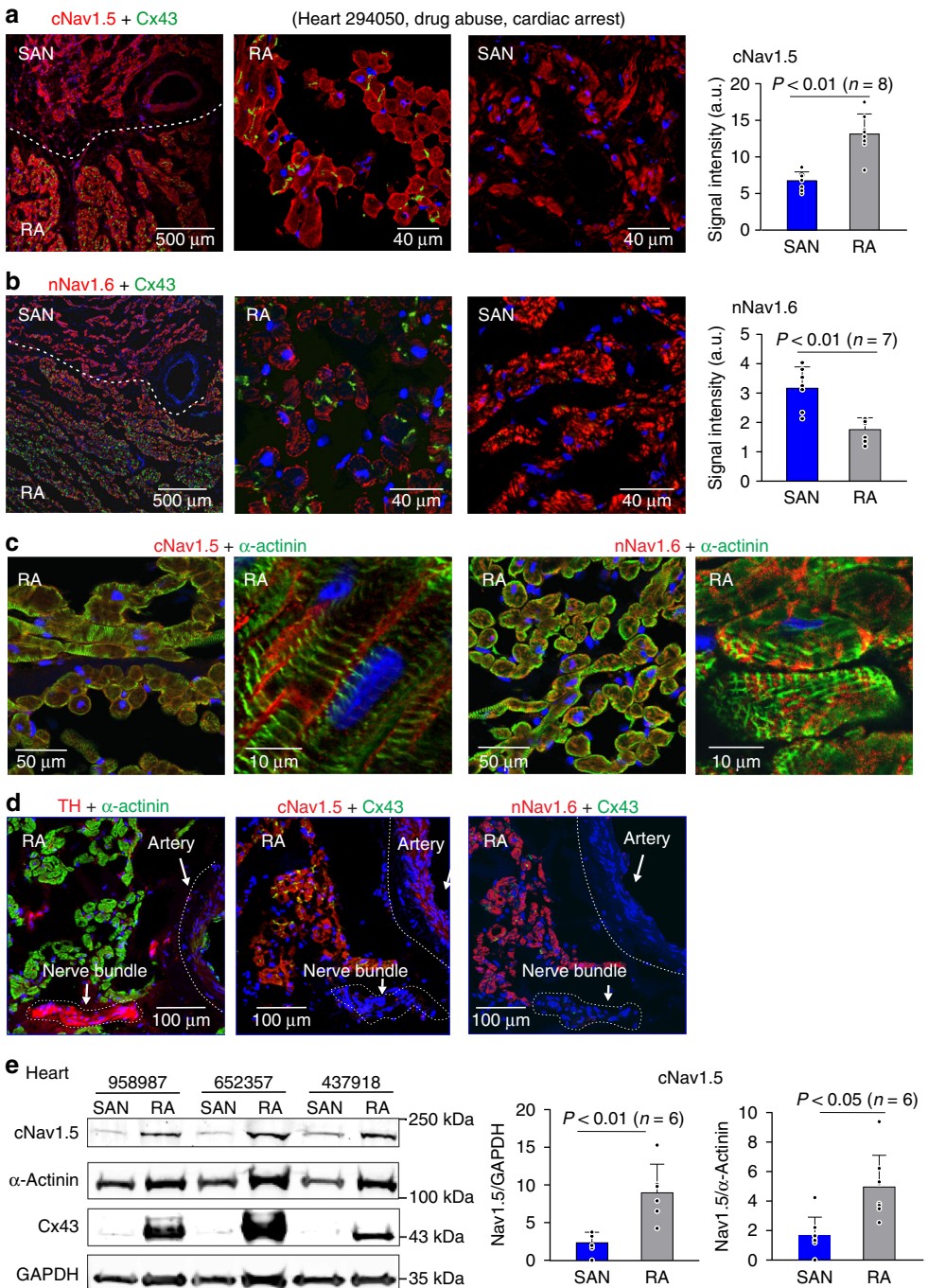

**Fig. 6 Protein distribution of cNav1.5 and nNav1.6 in the human SAN and RA. a** Left to right: immunofluorescence image showing double staining of cNav1.5 (red) and Cx43 (green) in human heart 294050 cryosection with sinoatrial node (SAN) and right atrial (RA) regions ($n = 8$) separated by white dotted line; magnification of image to show the distribution of cNav1.5 in the RA vs. the SAN; bar graph showing fluorescence signal intensity in the human SAN vs. RA. **b** Left to right: immunofluorescence image showing double staining of nNav1.6 (red) and Cx43 (green) in human heart cryosection with SAN and RA regions ($n = 7$) separated by white dotted line; magnification of image to show the distribution of Nav1.6 in the RA vs. the SAN; bar graph showing fluorescence signal intensity in the human SAN vs. RA. **c** Left to right: cNav1.5 (red) and α-actinin (green; staining cardiomyocytes) dual staining; nNav1.6 (red) and α-actinin (green) dual staining confirm the cardiomyocyte-specific localization of cNav1.5 and nNav1.6. **d** Serial sections staining cNav1.5, nNav1.6, Cx43, and a nerve bundle labeled by anti-tyrosine hydroxylase (TH: staining sympathetic nerves) show that nNav1.6 and cNav1.5 are predominantly found in the myocytes relative to nerve bundles. All presented images **a–d** were collected from human heart 294050. **e** Left: representative immunoblotting bands for cNav1.5, α-actinin (marker of cardiomyocytes), and Cx43. Right: summary data of immunoblotting results of cNav1.5 protein distribution in the human SAN ($n = 6$) and RA ($n = 6$), compared with GAPDH (middle) and α-actinin (right), respectively. Cx43 connexin-43, GAPDH glyceraldehyde 3-phosphate dehydrogenase. Data were represented in mean ± SD. For immunostaining, analysis was done using lme4 and emmeans packages in R 3.4.4. Predictors included Heart (treated as random effect) and Condition (fixed effect). Pairwise tests between Condition levels were adjusted using Tukey's method. Western blotting data analysis was done using two-sided $t$-test. Source data and uncropped versions of the western blot are provided as a Source Data file.

data may begin to explain the distinct functional contribution of nNav1.6 channels and correlate these findings with protein expression levels.

**nNav mRNA altered by HF and chronic alcohol consumption.** As every human heart studied is characterized by cardiac disease, aging, and/or comorbidities, we investigated whether mRNA profiles of any cNav and nNav are associated with specific disease and/or accompanying comorbidities and risk factors. To determine the contribution of these multiple factors, we screened the donor (non-failing) and HF hearts to identify those conditions (Supplementary Table 3). There were no significant differences in Nav mRNA levels associated with HTN, AF, coronary artery disease, gender, age, or illicit drug abuse in the hearts studied (Supplementary Tables 4 and 5). However, interestingly, history of chronic alcohol consumption was strongly associated with downregulation of multiple mRNA transcripts, both in the RA and SAN. *SCN2A*, *SCN8A*, and *SCN3B* transcripts were significantly downregulated in the RA and SAN of donor non-failing hearts with a history of chronic alcohol consumption (5/10 hearts) (Fig. 7a). Smoking history (5/10 hearts) did not show as strong effects as chronic alcohol consumption, with significant changes only in *SCN4B* in RA and SAN (Supplementary Fig. 8a). Besides chronic alcohol consumption, HF was also associated with significant downregulation of *SCN2A*, *SCN3A*, *SCN4A*, *SCN9A*, *SCN2B*, and *SCN4B* in RA but not in the SAN (Supplementary Table 3). Based on our finding that history of chronic alcohol consumption can independently modify several nNav transcripts, data were further analyzed excluding hearts with history of chronic alcohol consumption, to determine the specific effect of HF on Nav. Interestingly, four α-subunits and three β-subunits were significantly downregulated in the HF RA, including *SCN8A* encoding the nNav1.6 isoform, which was also significantly altered in the HF SAN (Fig. 7b). A positive correlation was also found between heart weight and *SCN11A* and *SCN1A* in RA and *SCN11A* and *SCN3B* in the SAN (Supplementary Fig. 8b). In addition, we also tested whether HF and chronic alcohol consumption are associated with differences in other major pacemaker ion channels in SAN but found no significant correlations (Supplementary Table 3).

**Computational modeling of Nav blockade in SAN-SACP-RA models.** We further explored the compartment-specific role of Nav isoforms using two-dimensional (2D) human SAN-SACP-RA computer models with and without HF characteristics (Figs. 8a, 9a), which incorporated distinctive structural features of human SAN and SACP[2,3] (Fig. 1). The 2D human SAN-SACP-RA computer models generated realistic characteristics (resting potentials, upstrokes, and action potential durations) of action potentials in SAN and RA regions, respectively (Fig. 8b). The computer model was validated by reproducing similar changes of SCL and SACT during 1–100 μM adenosine perfusion recorded in our previous optical mapping experiments[3]. Furthermore, the computer simulation revealed that impairments in SAN conduction during adenosine perfusion and $I_{Na}$ blockade were mainly due to inhibition of conduction in the SAN-SACP compartments (Fig. 8c–f), which directly correlate with functional mapping data (Figs. 1–4). Figure 8c–e represents the space–time plot of action potential profiles recorded from cells along the midline across the SAN-RA, showing continuous and stable action potential initiation and conduction at control condition, simulated 25 μM adenosine, or 20% $I_{Na}$ blockade in SAN (4% $I_{Na}$ blockade in RA). However, consistent with results from our optical mapping study, in contrast to solely adenosine or 20% $I_{Na}$ blockade, the combination of 25 μM adenosine and 20% $I_{Na}$

blockade led to 3:2 SAN exit block in SACP (Fig. 8f), suggesting that high levels of adenosine and HF-induced downregulation of multiple nNav in the RA could form a substrate for conduction blocks and rhythm failure. Under simulated HF conditions, 20% fibrosis was added to the SAN and SACP regions in the 2D human SAN computer models (Fig. 9a) and the resulting action potentials generated in SAN, SACP, and RA regions are displayed in Fig. 9b, respectively. Figure 9c shows that dose-dependent $I_{Na}$ inhibition by itself primarily slows SAN conduction and causes exit block at ~35% $I_{Na}$ blockade, with only ~10% SCL prolongation. However, the addition of even 10 μM adenosine lowers the threshold for SAN exit block, wherein complete exit block occurs at ~20% $I_{Na}$ blockade. Conditions simulating HF further slow SAN conduction and lower the threshold for both exit block and SAN arrest in the presence of adenosine or $I_{Na}$ blockade (Fig. 9d).

In the non-failing model, the lowest safety factor (SF) was found in the junction between SACP-RA at control conditions. Due to atrial influence, maximum diastolic potential decreased in SACP cells from −56 mV at SAN-SACP junction to −72 mV at SACP-RA junction, which significantly increased functional $I_{Na}$ across the SACP (Supplementary Fig. 9). Adenosine 25 μM induced hyperpolarization of ~1.5 mV across SAN-SACP-RA, which further increased functional $I_{Na}$ in SACP junction by 21% and SF by 5.8%, thereby maintaining source-sink balance necessary for conduction. Blocking $I_{Na}$ alone by 20% decreased SF by 14% in SACP junction. Under these conditions, adding 25 μM adenosine effects to the model produced the same hyperpolarization as adenosine alone, but as $I_{Na}$ was unavailable to compensate, a 3:2 SAN exit block was induced due to beat-to-beat decrease in SF to 1.003 during conduction and 0.904 during exit block. These results raised the possibility that the exit blocks could be caused by Nav channels in the atria rather than in the SAN-SACP. Hence, $I_{Na}$ was blocked separately, only in the SAN-SACP or in the RA. Results showed that when Nav channels are blocked only in the SAN-SACP component in the model, there are depressive effects on SCL, SACT, and thresholds of SAN conduction failure, which are similar to those of the control model, where Nav channels were blocked globally (Supplementary Fig. 10a, b). However, when Nav channels are blocked (up 40%) only in the RA, there is no SAN conduction failure clarifying that the exit blocks are mainly due to Nav channel blockade in the SAN-SACP (Supplementary Fig. 10c). The HF model had a similar pattern for SF across the SAN-SACP-RA with the lowest safety factor also found at the SACP-RA junction; however, a more pronounced effect on safety factor resulted from adenosine 25 μM = 26%, $I_{Na}$ block 20% = 25% and adenosine 25 μM + $I_{Na}$ block 20% = 59% (a safety factor of 0.7981). Furthermore, the computational human SAN models were able to reveal the mechanism for only exit blocks but the absence of entrance blocks during atrial pacing in functional mapping experiments when Nav were blocked (Figs. 3, 4). Computational analyses in control and HF human SAN models revealed that safety factor in SACP junction is direction-dependent and it is higher for entrance (~1.03) vs. exit conduction (~0.86) at the exit block conditions, which underlie the absence of entrance blocks during slow atrial pacing and $I_{Na}$ partial block (Supplementary Fig. 11). Importantly, as we have previously shown[3], the absence of entrance blocks in SACPs during atrial pacing further inhibited excitability in the SACP and promotes occurrence of post-pacing exit block (Supplementary Fig. 11b) and SAN reentrant arrhythmias[28], which can explain current experimental observations (Figs. 3, 4).

These results from computer simulations not only support our optical mapping findings, but also suggest that the importance of $I_{Na}$ in preserving SAN conduction is dose-dependently

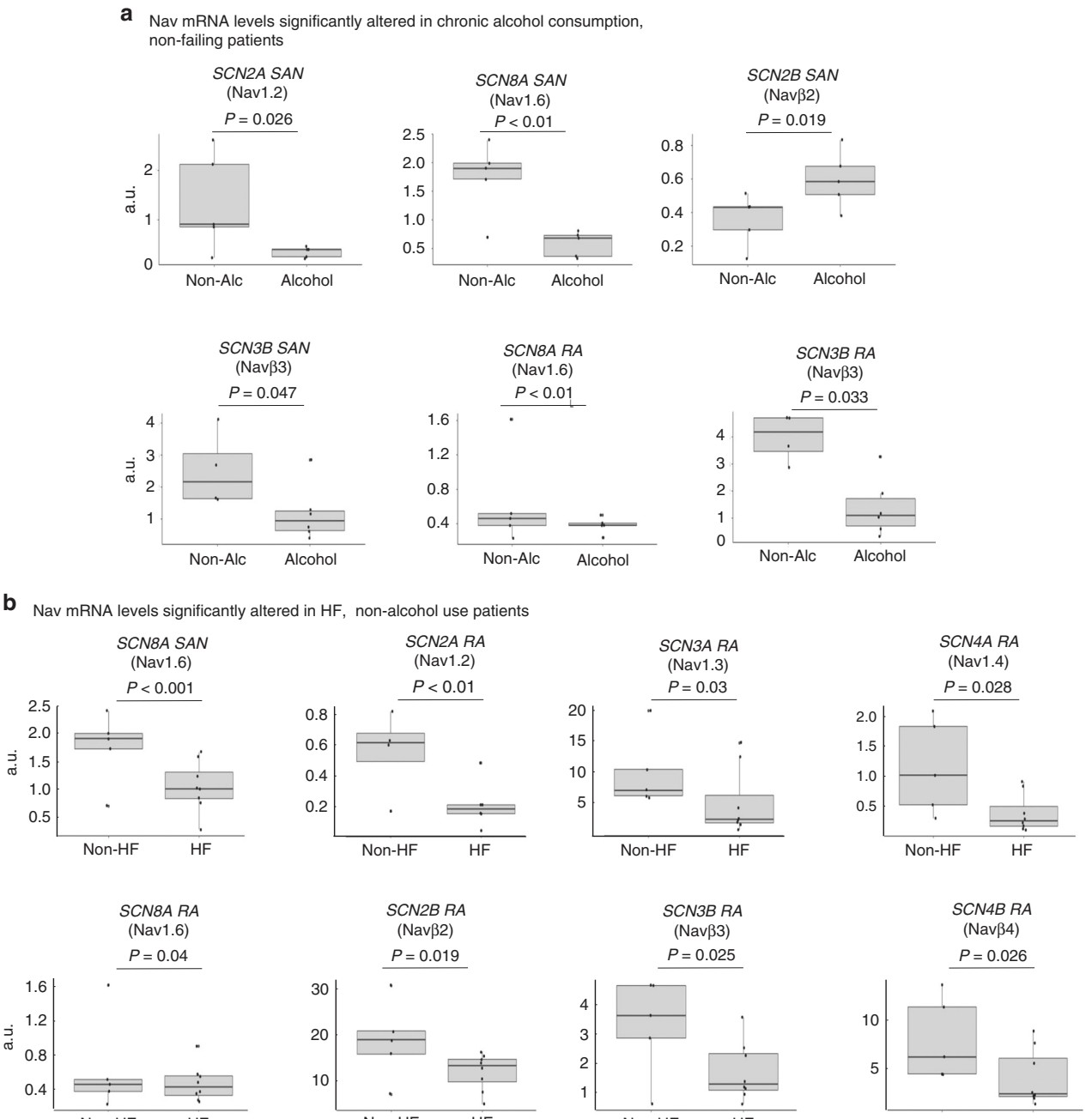

**Fig. 7 Nav subunits altered in chronic alcohol users and heart failure patients. a** Nav mRNA levels altered in human sinoatrial node (SAN) and right atria (RA) by chronic alcohol use in non-failing human hearts. Control $n = 5$, chronic alcohol use $n = 5$. **b** Nav mRNA altered in SAN and RA in heart failure (HF) non-alcohol users. Non-failing $n = 5$, HF $n = 8$. Data distributions are presented as box plots with dots as individual observations. The center line within the box represents the median, the lower and upper bounds of box represent the interquartile range, and the whiskers indicate the maximum and minimum sample values. Statistical analysis was done using mixed models in package lme4 with patient, heart weight, and indicator variables for alcohol, smoking, drug abuse, and HF. Patient was considered a random effect, others as fixed effects. Source data are provided as a Source Data file.

augmented by adenosine, resulting in SAN function failure due to exit blocks in SACPs and complete SAN automaticity arrest (Fig. 9c).

## Discussion
Here we report findings from near-infrared optical mapping that not only elucidate a previously unknown role for Nav channels[29] in the human SAN but also suggest that this functional contribution of Nav may be unique to the human SAN, compared

with the previously demonstrated roles of Nav in other small[30] and large[16] animal models. A major finding is that unlike cNav, nNav may predominantly contribute to SAN intranodal conduction, rather than atrial conduction. On the other hand, cNav play important roles in both SAN pacemaking and conduction, especially during adenosine or pacing-induced stress to prevent intranodal conduction failure. Furthermore, these functional observations are supported by higher expression of nNav (Nav1.1 and 1.6) and lower expression of cNav1.5 in human SAN cardiomyocytes vs. surrounding atrial tissue. Our data also show that

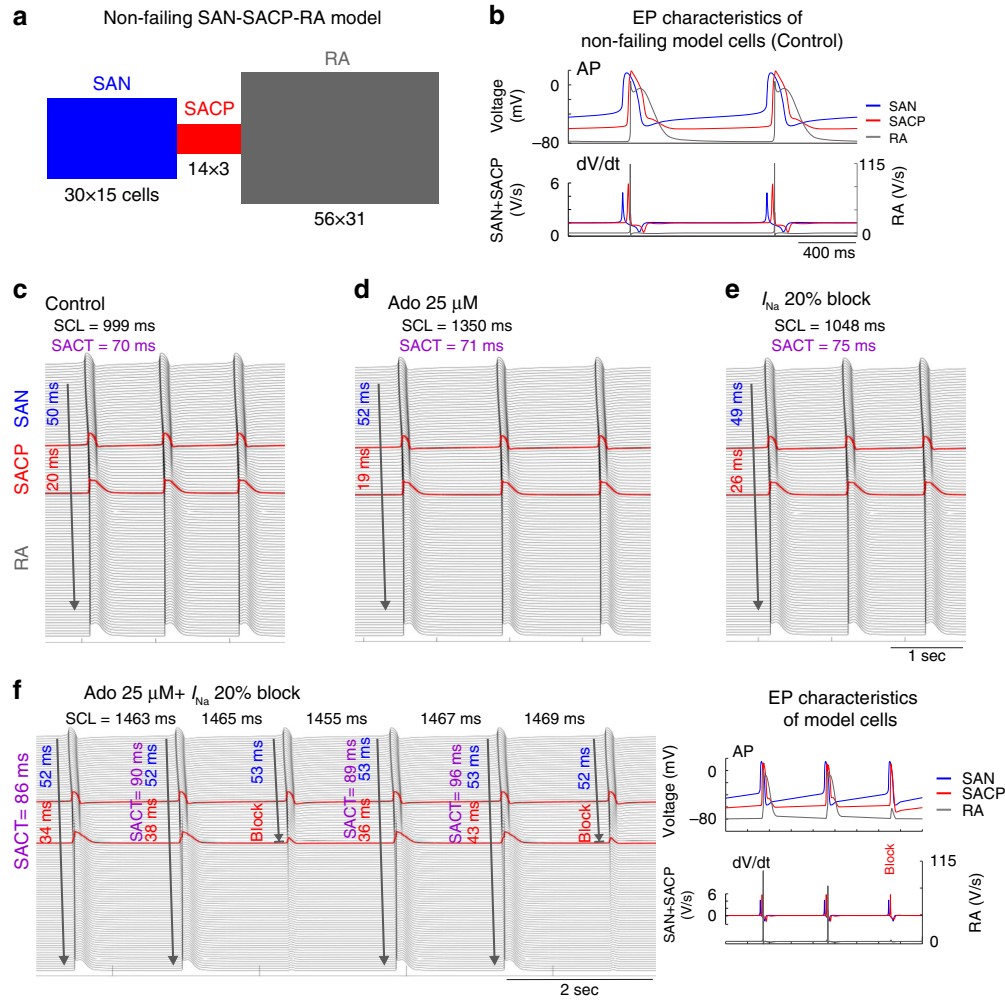

**Fig. 8 Human non-failing SAN computer simulations define Nav's protective role in SAN pacemaking and conduction. a** Geometry of 2D computer model. **b** Action potentials (AP) and their derivatives (d*V*/d*t*) show different electrophysiological (EP) characteristics of SAN, SACP, and RA cells in the current model. Propagation of APs along the middle axis of the 2D SAN-atrium are displayed from the top to bottom with time at control conditions (**c**), Ado (**d**), and sodium current ($I_{Na}$) 20% blockade (**e**). Blue and red numbers indicate the conduction time from SAN leading pacemaker through SAN and SACP, respectively. **f** Left: Ado plus $I_{Na}$ blockade reproduced prolonged SCL and SAN exit block. Right: representative APs and derivatives. Ado adenosine, RA right atria, SACP sinoatrial conduction pathway, SACT sinoatrial conduction time, SAN sinoatrial node.

several nNav transcripts were vulnerable to selective remodeling associated with HF, cardiac hypertrophy and modifying risk factors including history of smoking and chronic alcohol consumption. Biophysics-based computer modeling revealed compartment-specific mechanistic insights that suggest a protective role for $I_{Na}$ against rhythm failure within the human SAN pacemaker–conduction complex, especially in HF.

The human SAN is protected by multiple fail-safe mechanisms[3], which are critical to ensuring adequate cardiac performance as well as preventing SND and cardiac arrhythmias during pathophysiological conditions[13]. Currently, high-resolution, near-infrared optical mapping is the only mapping technique that is able to accurately identify intranodal conduction and hence capable of determining compartment-specific backup mechanisms[2,3]. Utilizing these high-resolution ex-vivo examinations was critical in our goal to uncover the unique roles of nNav and cNav within the human SAN. In the current study, blockade of both cNav and nNav was observed to directly inhibit SAN automaticity (SCL) at physiological conditions, manifesting an essential role for Nav in maintaining human SAN pacemaking. However, SACT proportionally increased more than SCL (SACT/ SCL ratio) during Nav blockade, which suggests that SAN

impairments due to Nav dysfunction may predominantly be due to depressed SAN conduction rather than automaticity. Importantly, Nav blockade increased beat-to-beat variability in SAN intranodal conduction and SAN reentrant arrhythmias (Fig. 2). These results establish a significant role for Nav channels in maintaining stable intranodal conduction and robust protection of human SAN rhythm, in both HF and non-failing hearts. In the light of our findings, we suggest that impaired intranodal conduction could underlie clinically observed symptoms of familial sick sinus syndrome including sinus bradycardia/arrest in some patients carrying *SCN5A* mutations[18,31].

Interestingly, although previous studies in animal models have identified a remarkable range of varying, species-specific roles for Nav isoforms in the SAN[14–16,30], our findings demonstrate that Nav channels may contribute very differently to human SAN pacemaking and conduction[15,16,32]. Studies have found that micromolar TTX can depress heart rate in adult mouse[33] and rabbit SAN preparations[34], providing evidence that cNav can contribute to SAN automaticity in some adult mammalian hearts. However, in contrast to the mouse SAN study[30], where nanomolar TTX impaired SAN automaticity but did not inhibit conduction, we found that nanomolar TTX impaired intranodal

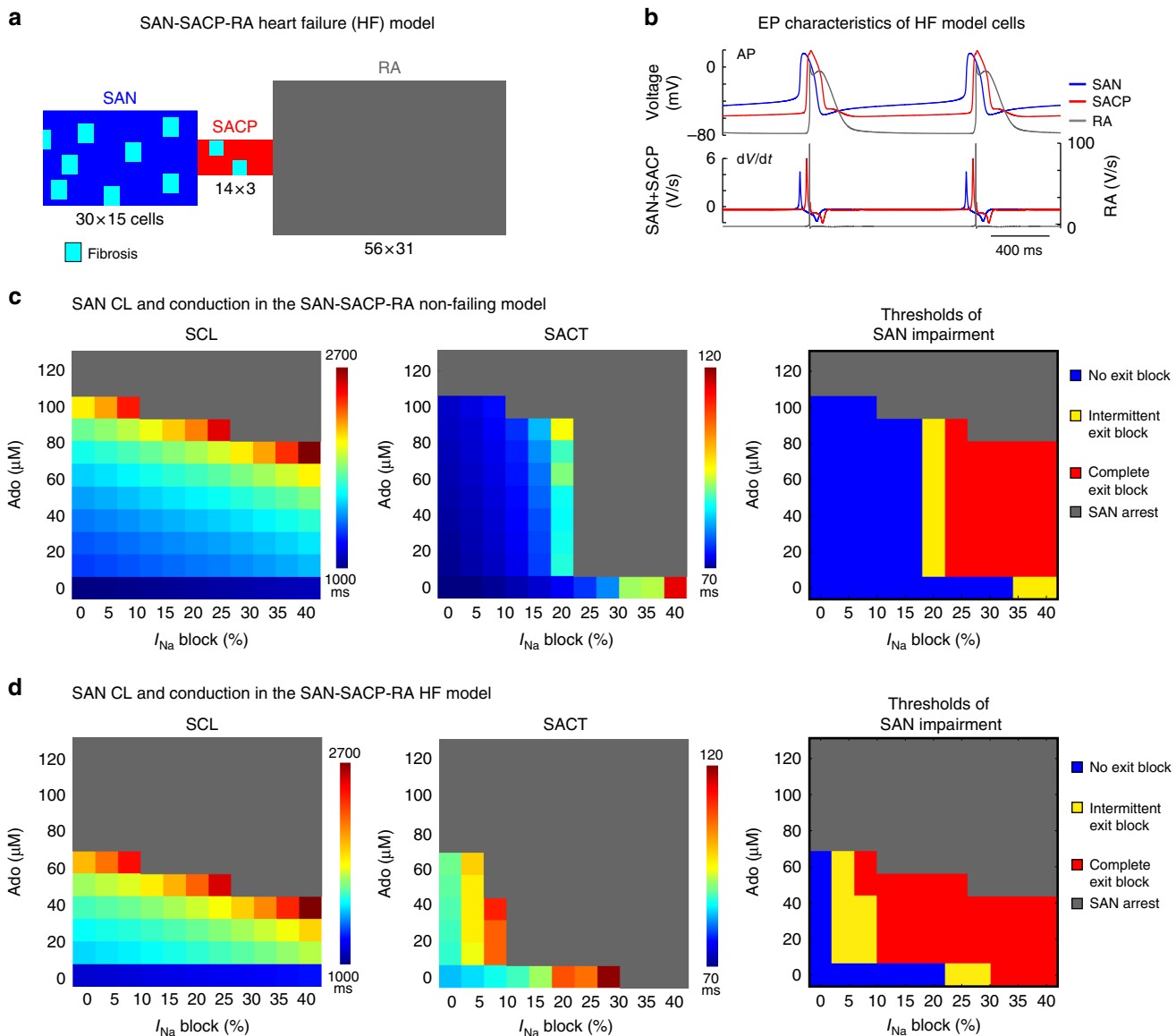

**Fig. 9 Heart failure increases SAN impairment caused by Nav blockade in computer simulations. a** Geometry of the 2D computer model with unexcitable fibrosis elements shown in light blue. **b** Action potentials (AP) and their derivatives (d$V$/d$t$) show different electrophysiologic (EP) characteristics of SAN, SACP, and RA cells in the current model. Table results displaying combinations of adenosine dose and the percentage of sodium current ($I_{Na}$) block in terms of cycle length, SACT, and threshold of SAN pacemaking and conduction impairment in control model (**c**) and HF model (**d**). Ado adenosine, HF heart failure, SACP sinoatrial conduction pathway, SACT sinoatrial conduction time, SAN, sinoatrial node, SCL sinus cycle length.

conduction (~250%) with negligible effects on SAN automaticity (~6%) and atrial conduction (~6%) at physiological conditions. Our data suggest that unlike other species, TTX-sensitive neuronal $I_{Na}$, particularly nNav1.6 in addition to cardiac $I_{Na}$, may particularly be important in maintaining intranodal conduction within the human SAN.

Increased levels of adenosine have been shown to cause SND during metabolic stress and HF[35,36]. In the human SAN, adenosine-induced inhibition of SAN pacemaking and conduction are mainly due to activation of A1R and outward potassium current ($I_{K,Ado/ACh}$)[3], which can significantly hyperpolarize SAN pacemaker cells[37]. Similarly, fast pacing or atrial arrhythmias could also increase the activation of outward $K^+$ currents and reduce $I_{Ca,L}$, which inhibit the excitability of the SACPs more than intranodal pacemaker compartments and lead to post-pacing SAN exit block in diseased hearts[2,38,39]. Therefore, we used both adenosine and fast atrial pacing to

simulate a pathological scenario to investigate the role of Nav channels in this context. Our findings show that Nav blockade significantly exacerbated the depressive effects of both adenosine and overdrive suppression resulting in intranodal conduction failure and SAN arrhythmia. These findings are in keeping with previous results from isolated SAN cells, suggesting that Nav could be preferentially activated when the cell is hyperpolarized[16,17], which may be important to counteract impaired SAN-SACP excitability during pathological challenges. Our data suggest that TTX-sensitive nNav could play a distinct protective role in human SAN conduction, especially during pathological conditions including adenosine-mediated hyperpolarization, when the balance between the source (electrical current/charge generated by the SAN and delivered through SACP) and sink (the current/charge required to activate the neighboring atrial myocardium) might be compromised (Figs. 3, 4).

Although direct experimental data confirming SACP-specific effects of $I_{Na}$ are lacking for the human SAN, our human compartment-specific SAN-SACP-RA computer model indeed demonstrates decreased maximal diastolic potential and increased $I_{Na}$ activation across the SACP toward the RA. This unidirectional increase in $I_{Na}$ activation within the SACP could help maintain the source-sink balance. Computer simulations further demonstrate that adenosine-induced hyperpolarization and $I_{Na}$ blockade can produce a synergistic negative effect, causing complete SAN exit block due to conduction failure in the SACP (Fig. 9 and Supplementary Figs. 10 and 11). Specific combinations of adenosine and $I_{Na}$ blockade reproduced the beat-to-beat conduction variability and intermittent SAN exit block observed in our optical mapping experiments.

Recent studies have shown that in addition to cNav, multiple nNav are expressed in the human heart[40]; however, their presence in the human SAN has never been investigated. Nav channels are composed of one pore-forming α-subunit associated with two different β-subunits[41] and can be categorized as TTX-resistant ($IC_{50} > 1 \mu M$) cardiac isoforms and TTX-sensitive ($IC_{50} < 60$ nM) neuronal channels[42]. In this study, mRNA transcripts of both nNav and cNav isoforms were detected, with varying distributions between the SAN and RA. In the SAN, among the TTX-resistant Navs (1.5, 1.8, and 1.9), only cNav1.5 was found at significant levels, indicating that the observed additional functional effects during high-dose TTX could be mediated mainly by cNav1.5. Importantly, the higher protein expression of nNav1.6 compared with cNav1.5 channels in SAN vs. RA may also explain the higher sensitivity of SAN to nanomolar TTX compared to surrounding atrial tissue. In fact, the specific nNav1.6 blocker depressed SAN function similar to the effects of 100 nM suggesting that the effect of TTX 100 nM in the human SAN studied may be partially mediated by nNav1.6; furthermore, higher ratio of nNav1.6 between SAN and RA was found in the heart that developed exit block with nNav1.6 blocker, compared with two other hearts that exhibited less sensitivity to nNav1.6 blocker. These results from the limited number of human hearts studied suggest that nNav1.6 plays an important role in preserving human SAN conduction potentially by preventing exit blocks. However, expression of other nNav isoforms and accessory/interacting proteins should also be investigated in future studies, to determine their mechanistic roles in SAN function.

Previous studies have shown that many cardiac diseases can cause remodeling of major ion channels[10-13] including HCN1 and HCN4, A1R, and L-type calcium channels. Indeed, we found significantly lower mRNA levels of almost all nNav isoforms in HF hearts, especially nNav1.6 in the SAN (Fig. 7). Interestingly, our data from the limited number of hearts studied show that nNav transcript levels in the human SAN and RA are also associated with other modifying risk factors including chronic alcohol consumption. As chronic alcohol consumption has been strongly associated with arrhythmias including ventricular tachycardia and AF[43], our data indicate that Nav channels must be studied in larger samples of hearts from chronic alcohol abuse patients to potentially reveal Nav-mediated arrhythmic mechanisms associated with chronic alcohol consumption. Interestingly, as shown in Fig. 4, our optical mapping data revealed that subsequent to nNav and partial cNav blockade, hearts with history of chronic alcohol consumption or HF were highly susceptible to SAN intranodal conduction disturbances leading to atrial beat-to-beat variability, SAN reentrant arrhythmias, and sinus arrest.

From a clinical standpoint, as adenosine's effects on the SAN are similar to those of vagal stimulation[37], our findings emphasize avoiding/limiting the use of drugs that may block TTX-sensitive $I_{Na}$, especially when vagal tone is high, or in HF[36] and AF[44] patients with high plasma levels of adenosine. Our results also identify the human SAN as a direct target for several clinical medications designed to block Nav channels, including Class I antiarrhythmic drugs that block Nav1.5, some anesthetics, and pain medications that block nNav. In fact, flecainide, a clinically prescribed Class I antiarrhythmic drug, has been reported to increase cSNRTi in SND patients[20] and to significantly reduce heart rate during exercise in patients with normal cardiac structure and SAN function[45]. The availability of Nav channels in the human SAN and RA could be an important backup mechanism to robustly protect SAN conduction and pacemaking, and prevent exit blocks and source-sink mismatch, in the context of multiple disease-induced conduction impairments[3,46].

Similar to other human studies, our data are also limited by the small number of ex-vivo human atrial preparations with varied pre-existing disease etiologies, which may not be representative of all human atria. As the data are from HF patients that have been treated long-term with several medications, disease-independent drug-induced effects could also have affected Nav transcript levels. Few HF hearts were available for optical mapping studies due to bicaval surgical dissection, which can impair SAN coronary artery perfusion. We were unable to confirm protein expression of all Nav isoforms transcripts detected in the mRNA studies due to the lack of specific antibodies suitable for human tissues. Furthermore, due low sample numbers we could not directly certify the correlation between Nav-related SAN pacemaking/conduction dysfunction and HF or alcohol consumption in optical mapping experiments. Additional studies should record $I_{Na}$ from isolated human SAN cardiomyocytes, in order to differentiate the specific contributions of cNav and nNav to cellular activation, relative to other ion currents including $I_f$ and $I_{Ca}$.

These findings provide insights to clarify the role of nNav and cNav isoforms in maintaining the robustness of the human SAN and establish Nav channels as important players essential to protect intranodal conduction and prevent rhythm failure.

## Methods

**Patient groups included in the study**. All human heart tissue research were approved by The Ohio State University Institutional Review Board and in compliance with all relevant ethical regulations. Informed consent for tissue collection was obtained from transplant patients and families of donors. Human hearts used in this study were de-identified and labeled with 6 digit random codes for reference. Hearts with intact SAN pacemaker complexes from transplant patients (with left ventricular hypertrophy, ischemic and non-ischemic HF, AF, and comorbidities including chronic HTN and diabetes) and human donor hearts (without history of HF and AF but with comorbidities including HTN, diabetes, and modifying risk factors including history of smoking, chronic alcohol consumption/abuse ($n = 34$, 19-68 y/o, details in Supplementary Tables 1 and 2) were obtained from The Ohio State University Cardiac Transplant Team or LifeLine of Ohio Organ Procurement Organization. Chronic Alcohol Consumption is defined as either abuse (> 7 drinks/week for women and > 14 drinks/week for men), as reported in documented medical records, or chronic consumption of moderate drinking (> 2 drinks/week for at least 10 years). Alcohol consumption was first collected from medical records when available and then from retrospective interviews with family members by LifeLine of Ohio Organ Procurement Organization for rejected donor hearts and drug abuse as listed in the patient's electronic health record. Expanded Materials and Methods are provided in the Supplementary Material.

**Near-infrared optical mapping and data analysis**. Conventional clinical electrode-based mapping systems are unable to record intramural conduction and activation patterns from within the 3D human SAN in patients, yet we were able to overcome these limitations by utilizing our recently validated high-resolution near-infrared optical mapping[3]. Human SAN preparations ($n = 14$) were coronary-perfused and stained with near-infrared dye di-4-ANBDQBS[3]. Regions of poor coronary perfusion/ischemia were excluded. To distinguish the function of different classes of Nav in the human SAN, two different doses of TTX (Abcam), 100 nM and 1–3 μM, were sequentially perfused through the coronary arteries. Lower dose, 100 nM TTX, selectively blocks majority of TTX-sensitive nNav subtypes (such as Nav1.1–1.3 and Nav1.6–1.7 with $IC_{50} \sim 10$ nM), but not TTX-resistant cNav1.5 with $IC_{50} \geq 1 \mu M$[42]. As the majority of nNav were already blocked, any further effects at the higher dose, TTX 1–3 μM, was attributed to blockade of TTX-resistant cNav1.5. Optical mapping was conducted with MiCAM Ultima-L CMOS cameras with resolution up to 330 μm$^2$ (100 × 100 pixels)[3]. Optical mapping

identified the leading pacemaker or earliest SAN depolarization, as well as earliest atrial activation sites, where activation exited to the atria through SACPs. SACT reports the time of activation propagation from the SAN leading pacemaker to the earliest atrial activation site. SAN activation patterns, SCL, SACTsr, or after atrial pacing (SACTppb), and corrected direct/indirect sinus node recovery time (cSNRT$d/i$) were measured during constant perfusion of Tyrode's solution at control ($n = 10$), and sequential perfusion of 100 nM ($n = 8$) and 1–3 μM ($n = 7$) TTX. Overdrive atrial pacing (CL = 500 ms, 400 ms, and 300 ms) and adenosine bolus (1 mL, 10–100 μM) were used to challenge the robustness of SAN pacemaking and conduction. RA CV was measured by RA pacing at 500 ms. In addition, a specific Nav1.6 blocker 4,9-Anhydrotetrodotoxin (30 nM)[27] was used in a subset of four hearts studied with the same protocols as for TTX above. Nav1.6 blocker was tested in five experiments with Heart 957855 tested before and after washout and addition of 1 nM isoproterenol (Supplementary Fig. 1 and Supplementary Table 1). When several drug protocols were studied, the first drug studied was washed out from SAN preparations and isoproterenol 1–10 nM may have been used to recover sinus rhythm to its baseline levels before the effect of Nav blockade was studied (Supplementary Tables 6 and 7).

**Molecular mapping and data analysis.** To molecularly map mRNA, protein expression, and distribution of nNav and cNav, pure SAN and surrounding atrial tissues were collected from 20 unmapped human hearts (non-failing = 10; failing = 10) guided by immunostaining and Masson's trichrome stain[3,4] (Supplementary Fig. 12). Total RNA and proteins were extracted separately from the SAN and surrounding atrial tissue[4,10]. qPCR was conducted in duplicates with QuantStudio 3 (Applied Biosystems), SYBR green (Qiagen), and QuantiTect primer assays (Qiagen; Supplementary Table 8). Comparative threshold cycle (Ct) was used to compare the relative abundance of mRNAs in the samples. Primary antibodies against Nav1.5 (custom-made in Dr Mohler's lab), Nav1.6 (Alomone), Connexin-43(Cx43), Glyceraldehyde 3-phosphate dehydrogenase, and α-actinin (Sigma-Aldrich) were used to quantify corresponding proteins by western blotting and immunostaining[3,4] (Supplementary Table 9). Uncropped versions of blots have been provided in Source Data file.

**Human compartment-specific SAN-SACP-RA computational model.** A biophysics-based 2D computational model was designed based on our study of human SAN structure[2] to simulate the interactions between SAN, SACP, and RA. The human SAN and SACP were modeled with the Fabbri et al. model[47], based on isolated human SAN pacemaker cell recordings[48], and adapted with optical mapping data from central SAN and SACP regions, respectively (Supplementary Table 10). The RA cells were modeled by using the original human atrial Courtemanche et al. cell model[49]. The ratio of $I_{Na}$ blockage in SAN vs. RA was set to 5:1. To incorporate the effects of adenosine into the cellular models, we utilized the acetylcholine (ACh)-activated K$^+$ current, $I_{K,ACh/Ado}$, previously used for the atria[49] and SAN models[47], equating ACh concentrations with those of adenosine based on our optical experiments. In HF SAN model (Fig. 9a), 20% reduction in $I_f$ current and heterogeneously seeded fibrosis with a size of $2 \times 2$ within SACP regions with a ratio of 1:4 to the normal SAN/SACP cells were implemented, as well as 20% and 5% reduction in the $I_{Na}$ current in SAN-SACP and atrial cells, respectively.

Conduction among SAN/atrial cells was modeled using a mono-domain equation and solved using a paralleled finite difference approach[50]. We used a spatial step of 0.03 mm and temporal step of 0.0025 ms in our solver. A forward Euler method[50] was used to solve the ordinary differential equations of cellular models. The conduction in the 2D tissue model attributable to intercellular electric coupling via gap junctions was modeled through the diffusion coefficient. In this model, we considered the regional differences in gap junctional coupling between the SAN, SACP, and RA tissues by setting the diffusion coefficient ratio to 2:14:100 in these three regions. The SF[51] was calculated to measure the success of propagation at each cell and is defined as the ratio of the total charge produced to the total charge consumed at that cell. If the ratio is less than 1, inefficient charge is produced for downstream activation and propagation will fail.

**Statistical analysis.** Data are presented as mean ± SD. All the statistical analyses were done in R 3.4.4 using packages lme4 and emmeans. The mixed models included Heart and Category (control, TTX 100 nM and TTX 1–3 μM) as predictors. Heart was treated as random effect and Category as fixed effect. Pairwise tests between condition levels were adjusted using Tukey's method. Quality of fit was monitored by visual inspection of residuals. Analysis of variance model was used to compare Categories. Analysis of the ratios for TTX 100 nM and TTX 1–3 μM categories compared with control was done using two-tailed $t$-test. Normality assumption was verified using Shapiro–Wilk test. Non-parametric data were analyzed with two-sided Wilcoxon's test. Analysis of exit block events was done using two-sided proportion test with Yates continuity correction. Association analysis of Nav isoform transcription level with diseases was done using mixed models in package lme4 with heart/patient, heart weight, and indicator variables for age, gender, smoking, chronic alcohol consumption, HF, AF, or HTN. Heart/patient was considered a random effect, others as fixed effects. $P$-values < 0.05 were considered significant.

**Reporting summary.** Further information on research design is available in the Nature Research Reporting Summary linked to this article.

## Data availability

Raw source data underlying all reported averages in graphs and charts, and uncropped versions of blots presented in the figures are available as a Source Data file. The source data underlying Figs. 1c, 2d, 3c, 5, 6 immunostaining, 6 western blotting, Table 1, and Supplementary Figs. 5, 7 are found in the source data file. All data are presented within the manuscript, the online data supplemental file, or are available upon reasonable request to the corresponding author.

## Code availability

The ionic model utilized[47] is freely available from the repository CellML (https://www.cellml.org/). Link to computer model source code, the test and readme files are provided here: https://github.com/Charcol97/Fabbri_HumanSAN_OSU

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

## Acknowledgements

This work was supported by NIH HL115580 and HL135109, American Heart Association Grant in Aid #16GRNT31010036 (V.V.F.), NIH T32HL134616, NIH F30HL142179 (B.J.H.), HL114940 (B.J.B.), and by funding from the Dorothy M. Davis Heart and Lung Research Institute. We thank the Lifeline of Ohio Organ Procurement Organization and the Division of Cardiac Surgery at The OSU Wexner Medical Center for providing the explanted hearts. We thank Mr Farbod Fazlollahi, Ms Salome Kiduko, and Ms Kyra Peczkowski for their help with tissue processing.

## Author contributions

This project was designed by V.V.F., with the methodology developed by N.L., B.J.H., A.K., and V.V.F. The investigations were performed by N.L., E.J.A., B.J.H., S.H.A., K.M.H., G.R., B.W., N.A.M., and V.V.F. Simulation was performed by R.S. and J.Z. Data analysis and statistic were performed by N.L., S.H.A., E.J.A., P.J.W., and S.Z. Supervision of the project was provided by S.G., P.M.L.J., B.J.B., F.A., P.J.M., J.D.H., H.D., and V.V.F. The original draft was written by N.L., A.K., and V.V.F., and revised by N.L., A.K., E.J.A., B.J.H., K.M.H., and V.V.F.

## Competing interests

The authors declare no competing interests
