## [Peer Review File · Nature Communications]

Reviewers' comments:

Reviewer #1 (Remarks to the Author):

The study by Li et al reports on a newly identified role of voltage-gated sodium channels in human sino-atrial node pacemaking function. Using optical imaging, the authors demonstrate that impairment of Nav channels perturbs SAN pacemaking, can lead to changes in intranodal conduction and in some cases can lead to failed conduction into the right atrium. The functional data is further supported by molecular studies and immunostaining on tissue slices. A correlation with co-morbidities such as heart failure, and extrinsic factors such as alcohol consumption is also suggested. The results are novel and the work on human tissue samples is both impressive and important.

General comments:

1. Although the authors show convincing data on the role of Nav current in human pacemaking, no mention is made to the other known mechanisms of pacemaking, including If and calcium homeostasis. They are completely absent from the introduction and discussion and not taken into account in the experimental protocol. It is therefore unclear what the relative role is of Nav in a pathological context, especially since these other mechanisms will also likely be affected. Furthermore, adenosine and overdrive pacing used here to challenge SAN function, will also directly impact these other components of the pacemaker function.
2. The presented cases of conduction failure into the right atrium (experimentally in Figure 4 and computationally in Figure 8) show some degree of SAN pacemaking disturbance, but it could be argued that the actual conduction failure is due to impairment of sodium channels in the right atrium and not in the SACP.

Specific comments:

1. The authors claim at several places that their approach using high-resolution near-infrared optical imaging allows to reveal 3D human intranodal SAN pacemaking and conduction. Although the use of near-infrared dyes probably helps in obtaining acceptable signal-to-noise ratios from epicardial recordings in such tissue preparations, the method used here remains 2D epi-fluorescence and not 3D optical imaging. Furthermore, the increased penetration of red light into tissue due to reduced scattering and absorption, in fact significantly decreases effective spatial resolution, over conventional short wavelength imaging. The authors should therefore tone down their claims, as no 3D data is presented here and optical action potentials are averaged over significant tissue volumes.
2. In the first paragraph of the results it is stated that the tissue preparations exhibit stable intrinsic sinus rhythm within the range of in vivo measurements during autonomic blockade. Yet, in the online methods the authors mention isoproterenol was necessary to recover sinus rhythm. What

was the sinus rhythm without isoprotorenol? What are the effects on isoproterenol at the used concentrations on the baseline electrophysiological properties and sodium channel function?

3. The authors follow a specific protocol with serial addition of drugs and gradual increase in TTX concentration. From the online methods it can be inferred that the total experimental (perfusion) time can be up to 2 hours or more. It is known that under Tyrode perfusion, tissue properties will drift with time (conduction properties included). It would therefore be useful to provide control data in order to exclude any effects of time on the presented results (or alternatively randomize the drug protocol).

4. Although this reviewer appreciates the challenges associated with working on human samples, the correlation between Nav-related SAN pacemaking dysfunction and HF or alcohol consumption, can at a functional level (optical mapping data) not be certified with the current n (2 HF vs 8 non-HF, 3 "alcoholic" vs 7 "non-alcoholic").

Reviewer #2 (Remarks to the Author):

In this manuscript, Li and colleagues present the results of experiments aimed at examining sinoatrial node (SAN) functioning in non-failing human hearts and at exploring the mechanisms contributing to SAN dysfunction in failing human hearts. The authors state, in the abstract (opening sentence), that the mechanisms underlying human SAN dysfunction are poorly understood and that it is unclear whether robust human SAN excitability requires voltage-gated sodium (Nav) channels. These are the two topics that the authors then set out to address using a combination of optical mapping, pharmacological and molecular approaches. Interestingly, however, the authors do suggest somewhat modified (and larger) objectives in the introduction (on page 5, paragraph 2, lines 84-92) where they state that "the primary goal of the current study is to determine the existence and specific role of [neuronal Nav] nNav and [cardiac Nav] cNav isoforms in human SAN pacemaking and intranodal conduction and to reveal their disease-induced alterations in explanted human hearts, with/without [heart failure] HF and associated comorbidities, including atrial fibrillation, hypertension (HTN), and modifying risk factors such as smoking, chronic alcohol consumption, and drug abuse. As presented, therefore, the scope of the studies undertaken seems, from the outset, to be very large.

Functional, optical mapping, studies, as well as molecular and biochemical analyses of Nav channel subunit expression in human SAN, compared with human right atria (RA), are presented. The approaches used are reasonable and potentially could be powerful, providing insights into the relationships between Nav channel expression patterns and functioning. The present manuscript falls short of doing this, however, as the numbers of samples/hearts analyzed, particularly in the optimal mapping section of the study/manuscript, are very small and no attempt is made to correlate the various data sets (i.e., the mapping, the molecular and the biochemical data). It is

certainly appreciated that this sort of multiscale analyses is difficult. It also seems reasonable to assert, however, that to make a significant impact on the field, this would need to be done. In the absence of this sort of rigorous and robust experimental design, the experimental observations really are just that and they will remain as such until more robust experiments and analyses are completed.

As it stands, this manuscript presents two important findings, i.e., that Nav channels are functionally expressed in human SAN and, in addition that both cNav and nNav are expressed and functional. The molecular data presented suggest that the transcript expression levels of multiple Nav channel alpha and beta subunits are affected (and differentially so) by various conditions, from chronic alcohol consumption to heart failure. The functional consequences of these differential changes in transcript levels is not clear and, importantly, was not explored directly. The authors do present some molecular/computational modeling to begin to address the relationships between the molecular, cellular and circuit components and the role(s) of Nav channels in the SAN and RA. The findings presented are interesting, although interpreting the functional impact/significance is limited by the lack of experimental testing and validations of the model. In addition, the modeling efforts are limited in that the exclusive focus of the simulations presented was Nav channels; other ion channels, including ion channels likely to be differentially expressed in SAN and RA, appear not to be considered.

The authors might well be correct in their interpretations of the functional implications of their findings. There are, however, a number of concerns with the data presented and with the authors analyses and interpretations of the experimental data acquired that limit enthusiasm for this work as presented. Some of these concerns are enumerated below.

Major concerns:

1. As noted above, the initial studies presented are the optical mapping studies and the authors indicate that have mapped eight ($n = 8$) non-failing and two ($n = 2$) failing human hearts. On reading the text, it is unclear why mapping was done on only these 10 hearts when more were clearly available (and used for molecular and biochemical analyses). It appears, from the presentation of the optical mapping data that the data from these 10 hearts were combined and analyzed together, although no explanation or rationale is provided. Focusing on the main point of the paper, however, the authors do present data showing that perfusion of tetrodotoxin (TTX), a highly selective Nav channel blocker affects cycle length and conduction time, consistent with effects on conduction velocity, in isolated human SAN, at concentrations in the range of 100 nM to 3 μ M. Dramatic effects of TTX on conduction were also seen in preparations subjected to rapid atrial pacing or pre-treated with adenosine. It is unclear whether there are any detectable differences (in TTX sensitivity)

between failing and non-failing hearts as, again, this point seems not to have been addressed (likely owing to the very small sample numbers). The rationale for the small sample numbers and for combining the functional (i.e., mapping) data acquired from non-failing and failing hearts remains unclear and unexplained.

2. It is unclear why the authors didn't design and complete experiments in which neuronal and cardiac channels were selectively blocked, and the functional effects of these select manipulations were determined?

3. The authors do go on to examine the mRNA and protein expression levels of several Nav channel (alpha and beta) subunits using quantitative RT-PCR, immunostaining and western blot analyses of SAN tissue, compared with surrounding right atrial (RA) tissue (in the same hearts). It is unclear from the data presentation, however, how the molecular data relate to the functional data. It is similarly unclear how the biochemical data relate to the functional data, although this concern can be minimized by simply stating that the biochemical data should not be considered (i.e., interpreted) at all here, as only two Nav channel alpha subunits were examined.

4. It is unclear why the authors restricted their molecular analyses to Nav alpha and beta subunits. Why, for example, weren't the expression levels of other Nav channel accessory/ interacting protein subunits also examined? And, why weren't other channels (and receptors) that are likely differentially expressed in human SAN and RA also considered?

5. One wonders why the authors did not use more comprehensive molecular strategies/ approaches to analyze the transcriptomes of their human SAN and RA tissues (i.e., rather than using RT-PCR to analyze one transcript at a time)?

6. One similarly wonders why the authors did not attempt to correlate their individual data sets...i.e., the functional imaging and transcript expression data collected for the individual human hearts studied?

7. It seems that the computer simulations of human SAN function and dysfunction are based on more assumptions, than actual data, and are, as a result, somewhat misleading.

Additional Specific Comments:

1. It is unclear what the authors mean (in the abstract) by noting that they have used "high-resolution" molecular mapping?

2. The terms "conduction catastrophe" and "electrical stress" (also used in the abstract) are unlikely to be understood (or interpretable) by non-experts. The authors need to make the experiments, results and conclusions presented accessible to a general audience.

3. In presenting the analyses of their molecular (Nav channel subunit transcript expression) data), the authors repeatedly refer to “changes” in channel subunit expression levels. But the differences observed between different samples are not “changes”, they are simply “differences” as the same sample was not measured more than once, as would be needed to describe something as a “change”.

4. The Discussion section of this manuscript is very long (6+ pages), particularly given the limited scope of the work, and could be reduced by 50% with no effect on the impact or import of the paper.

Reviewer #3 (Remarks to the Author):

This is a well executed paper that helps to address an important scientific and clinical question, namely the role of the sodium current in cardiac pacemaking and why patients with sodium channel mutations may present with sinus node dysfunction.

I have some minor comments:

- The term 'conduction catastrophe' is overly dramatic and lacks any explanation of the underlying physiology. I would encourage the use of a less emotive and more descriptive term.

- The sentence "Many patients with symptomatic SND are found to harbor loss-of-function mutations in the SCN5A gene" should read "some patients". SCN5A mutations are a recognised but rare cause of SND

- Results: Heart rates of 56-116 bpm are stated to be in the normal range of intrinsic heart rates for humans this is not correct. Reference should be made to the paper "Jose & Collison, The normal range and determinants of the intrinsic heart rate in man, 1970" which should be considered the gold standard for intrinsic heart rates. Thus some of the hearts in this study have an intrinsic heart rate below the normal intrinsic rate. This does not detract from the scientific value of the paper.

- Results: Control n=8 but TTX is n=7 / 6. Comment should be made as to why some of the control hearts did not go on to be included in the TTX dataset.

- Results: "prolongation of SCL by cNav blockade was primarily due to depression of SAN conduction rather than automaticity." This does not make any sense in terms of the mechanisms of SAN automaticity. The authors should reference any data showing that SAN conduction velocity can alter automaticity or state that the mechanism by which this occurs is not clear.

- Results / methods: "Chronic Alcohol Consumption" – does this mean alcohol 'abuse' or drinking any alcohol at all. How was this data gathered for the clinical database? Was it from patients or relatives? Retrospective or prospective? How many units of alcohol, how long determines 'chronic' etc? More detail is needed for the methods

- Results: Data is presented regarding SAN re-entry. This is interesting but the phenomenon of SAN re-entry is controversial. There is not sufficient data shown to allow the reader to make an interpretation of the maps to confirm re-entry is occurring. More detail should be given - eg videos showing the activation maps (including the atrial conduction), or at the very least detail of the interaction between the atrium and SAN during this phenomenon, perhaps it would be helpful if this data were 'zoomed in' on the area of re-entry to make a more convincing figure. I would also be interested to see the activation of the SAN and the exit to the atrium from the SAN beat that is initiated by re-entry. Lastly in this regard, can the authors identify any critical features that lead to the conditions for re-entry? For example conduction time from exit to entry point? Conduction velocity in the atrium? etc

- Discussion: It is interesting that TTX causes exit block but not entrance block. Can the authors speculate on why?

Response to Reviewers:

We thank all Reviewers for their overall positive evaluation of our study and their helpful criticisms and suggestions. We believe that the revisions made in all three main sections (functional mapping, molecular analyses, and computer modeling) as per the editor's and reviewers' suggestions have significantly improved our manuscript and clarified our data presentation. To expedite the review, we present our revisions to the manuscript in blue font. We have made the following major changes to improve the paper:

1. As requested by Reviewer 2 and 3, we changed the term “conduction catastrophe” to “conduction failure” and changed the title to “Impaired neuronal sodium channels cause intranodal conduction blocks and arrhythmias in failing human sinoatrial node”
2. As requested by the editor and Reviewer 2, we have conducted additional human SAN optical mapping experiments wherein we tested a new selective Nav1.6 blocker to reveal the distinct effects of selective nNav isoform blockade on SAN function failure during autonomic stimulation and overdrive. The results have strengthened our study by confirming the central hypothesis that nNav uniquely contribute to human SAN conduction function and we have included new data to **Figure 3 Panel C and Online Figure I, III-V**.
3. Furthermore, in order to provide the requested direct correlations between functional and molecular data, we have quantified cNav1.5 and nNav1.6 protein expression from immunostained SAN and RA tissue sections collected from new hearts, after optical mapping with specific nNav1.6 blocker (**Online Figure VII**).
4. As requested by Reviewer 2, we have also conducted additional experiments on the same samples presented in the earlier version (n=20) to quantify the transcript levels of additional protein targets including HCN1, A1R, and calcium channel isoforms in order to present a comprehensive evaluation of other proteins and channels involved in automaticity and conduction, in addition to Nav channel isoforms that were presented in the earlier version.
5. As requested by Reviewer 2, we validated our human SAN computer model by functional and molecular experimental data. Importantly, with the validated computational human SAN models, we have revealed the contribution of selective Nav blockade in SAN-SACP vs atria in SAN dysfunction (Reviewer 1). We have also addressed the mechanism that could lead to exit blocks without entrance blocks

during Nav blockade (Reviewer 3), due to direction-dependent safety factors of SAN conduction pathway.

6. We have updated most of our Figures and Tables and added six new Figures (**Online Figures I, III, IV, VII, X, and XI**) and two new online movies. As requested by Reviewer 2, we have reduced the length of the discussion.

Below, please find our detailed responses to the specific comments made by each Reviewer.

Response to Reviewer #1:

The study by Li et al reports on a newly identified role of voltage-gated sodium channels in human sino-atrial node pacemaking function. Using optical imaging, the authors demonstrate that impairment of Nav channels perturbs SAN pacemaking, can lead to changes in intranodal conduction and in some cases can lead to failed conduction into the right atrium. The functional data is further supported by molecular studies and immunostaining on tissue slices. A correlation with co-morbidities such as heart failure, and extrinsic factors such as alcohol consumption is also suggested. The results are novel and the work on human tissue samples is both impressive and important.

We thank Reviewer 1 for his/her valuable comments which helped us to considerably improve our manuscript and data presentation.

General comments:

1. Although the authors show convincing data on the role of Nav current in human pacemaking, no mention is made to the other known mechanisms of pacemaking, including If and calcium homeostasis. They are completely absent from the introduction and discussion and not taken into account in the experimental protocol. It is therefore unclear what the relative role is of Nav in a pathological context, especially since these other mechanisms will also likely be affected. Furthermore, adenosine and overdrive pacing used here to challenge SAN function, will also directly impact these other components of the pacemaker function.

Response: Thank you for this very pertinent question. We agree with the reviewer that the mechanism of SND is mediated by several interacting ion channels and signaling mechanisms and not limited to only Nav channels. This first-time study was undertaken to clarify the

functional role of both cNav and nNav channels specifically in the human SAN. Our results from the limited number of human hearts studied, especially the new experiments added using neuronal nNav1.6 channel specific blockade, confirm the functional contribution of nNav to maintaining SAN pacemaking and conduction. Furthermore, our results also show that Nav blockade causes an SND phenotype during challenging conditions, including adenosine and overdrive pacing, which could simulate pathological contexts in remodeled human hearts. These results clearly warrant further in-depth investigations to reveal how Nav channels contribute to pacemaking and conduction relative to the other components of both, ion and calcium channels.

We agree with the reviewer that adenosine and overdrive pacing can depress SAN function by different mechanisms including inhibition of I_f and calcium currents. In fact, we presented our data on Nav blockade in paired comparison within the same heart to control inter-heart differences in other ion channels. Additionally as suggested, in order to present a more comprehensive picture of the pacemaking mechanisms (rather than focusing solely on Nav), we have now added I_f and calcium channel components in the introduction and discussion.

Furthermore, we have now included new *in-silico* studies using our human SAN-specific computer models wherein key ion channel currents have been altered to simulate pathological remodeling. Our results clearly show that while Nav blockade by itself can slow SAN conduction, additional factors, including decreased I_f and I_{Na} , and increased fibrosis) can exacerbate these depressive effects indicating, as the reviewer suggested, that Nav function can be modulated by other components of the remodeled SAN and RA (**revised Figure 9 and new Online Figures X and XI**). Based on these interesting findings we plan to address these complex interactions in the future with more improved and heart-specific 3D computer models as to where and how Nav channel activity fits into normal and disease scenarios, factoring-in more ion-currents, in order to better understand the role of Nav channels.

2. The presented cases of conduction failure into the right atrium (experimentally in Figure 4 and computationally in Figure 8) show some degree of SAN pacemaking disturbance, but it could be argued that the actual conduction failure is due to impairment of sodium channels in the right atrium and not in the SACP.

Response: The reviewer raises a very relevant question. In order to test this possibility, we conducted additional human SAN computational simulations in which Nav channels were

blocked either only in the SAN-SACP, or in the atrial cells vs both simultaneously as we previously reported in **Figure 9**. The new **Online Figure XI** shows that when Nav channels are blocked only in the SAN-SACP component in the model, there are depressive effects on SCL, SACT, and thresholds of sinoatrial conduction failure which are similar to previous **Figure 9**, where Nav channels were blocked globally. However, when Nav channels are blocked only in the atrial cells of the model, there is no SAN conduction failure. The new data have been added to Results section page 12 and **Online Figure XI**.

Specific comments:

1. The authors claim at several places that their approach using high-resolution near-infrared optical imaging allows to reveal 3D human intranodal SAN pacemaking and conduction. Although the use of near-infrared dyes probably helps in obtaining acceptable signal-to-noise ratios from epicardial recordings in such tissue preparations, the method used here remains 2D epi-fluorescence and not 3D optical imaging. Furthermore, the increased penetration of red light into tissue due to reduced scattering and absorption, in fact significantly decreases effective spatial resolution, over conventional short wavelength imaging. The authors should therefore tone down their claims, as no 3D data is presented here and optical action potentials are averaged over significant tissue volumes.

Response: Thank you. As suggested, we have changed the sentence to “We employed high-resolution near-infrared sub-surface optical mapping, the only approach currently able to reveal human intranodal SAN pacemaking and conduction”

2. In the first paragraph of the results it is stated that the tissue preparations exhibit stable intrinsic sinus rhythm within the range of in vivo measurements during autonomic blockade. Yet, in the online methods the authors mention isoproterenol was necessary to recover sinus rhythm. What was the sinus rhythm without isoproterenol? What are the effects on isoproterenol at the used concentrations on the baseline electrophysiological properties and sodium channel function?

Response: We thank the reviewer for this question and apologies for the confusion. Isoproterenol bolus (0.2 mL of 1 μ M) was used in some experiments to help recover sinus rhythm during first 10-40 minutes of the experiments when SAN preparations were acclimatized to the warm perfusion bath, from the ice-cold cardioplegic solutions, and warmed up from 4 to 37° C. During these first minutes of coronary perfusion, automaticity could be depressed in SAN

preparations and adding bolus of isoproterenol quickly restores stable pacemaker and conduction function, which also facilitates contraction necessary for rapid washout of cardioplegic solution from interstitial compartments. Additionally, if several drug protocols were studied, then the first drug studied was washed out from SAN preparations, and 1-10 nM isoproterenol may have been used to recover sinus rhythm to baseline levels before the effect of Nav blockade was studied. We have added this information to the Extended Methods and additional **Online Table VIII and IX** to the supplemental data to compare the effect of Nav channel blockade on SCL, SACT, and atrial conduction velocity with and without isoproterenol. Importantly, isoproterenol had a predictable effect, which decreased SCL, but did not have any appreciable effect on functional changes due to Nav blockade.

3. The authors follow a specific protocol with serial addition of drugs and gradual increase in TTX concentration. From the online methods it can be inferred that the total experimental (perfusion) time can be up to 2 hours or more. It is known that under Tyrode perfusion, tissue properties will drift with time (conduction properties included). It would therefore be useful to provide control data in order to exclude any effects of time on the presented results (or alternatively randomize the drug protocol).

Response: We agree with the reviewer's statement. Unfortunately, due to the non-selective nature of high-dose TTX, we cannot randomize the drug protocol without extended periods of time between conditions to ensure adequate washout. The reviewer is correct that the protocol took roughly 2 hours to complete, with the first ~45 minutes constituting the control or baseline condition. We have now calculated the natural drift over these 45 minutes in sinus cycle length, sinus node recovery time, sinoatrial conduction time, and right atrial conduction velocity to exemplify the net effect attributable to intervention with TTX or selective nNav1.6 blocker. In general, we only observed a drift in sinus cycle length of ~5% per hour without a drift in atrial conduction velocity, which suggests that changes in automaticity and conduction seen with low and high dose TTX, respectively, were significant and not due to changes in tissue properties during mapping experiments. This new data can be found in new **Online Figure III** and in Results pages 6-7.

4. Although this reviewer appreciates the challenges associated with working on human samples, the correlation between Nav-related SAN pacemaking dysfunction and HF or alcohol

consumption, can at a functional level (optical mapping data) not be certified with the current n (2 HF vs 8 non-HF, 3 "alcoholic" vs 7 "non-alcoholic").

Response: We agree with the reviewer that we cannot arrive at statistically significant claims on functional differences between HF, chronic alcohol consumption, and non-chronic alcohol consumers due to small sample numbers. To better show the functional correlation of Nav-related SAN pacemaking dysfunction and HF or chronic alcohol consumption, in the revised version, we added new **Online Figure I** to show the response of individual heart to adenosine, atrial pacing (SNRT) with and without neuronal and cardiac Nav blockades. Furthermore, we clarified in the Limitations section that we could not directly certify the correlation between Nav-related SAN pacemaking/conduction dysfunction and HF or alcohol consumption in optical mapping experiments due low sample numbers.

Response to Reviewer #2:

Reviewer #2 (Remarks to the Author):

In this manuscript, Li and colleagues present the results of experiments aimed at examining sinoatrial node (SAN) functioning in non-failing human hearts and at exploring the mechanisms contributing to SAN dysfunction in failing human hearts. The authors state, in the abstract (opening sentence), that the mechanisms underlying human SAN dysfunction are poorly understood and that it is unclear whether robust human SAN excitability requires voltage-gated sodium (Nav) channels. These are the two topics that the authors then set out to address using a combination of optical mapping, pharmacological and molecular approaches. Interestingly, however, the authors do suggest somewhat modified (and larger) objectives in the introduction (on page 5, paragraph 2, lines 84-92) where they state that “the primary goal of the current study is to determine the existence and specific role of [neuronal Nav] nNav and [cardiac Nav] cNav isoforms in human SAN pacemaking and intranodal conduction and to reveal their disease-induced alterations in explanted human hearts, with/without [heart failure] HF and associated comorbidities, including atrial fibrillation, hypertension (HTN), and modifying risk factors such as smoking, chronic alcohol consumption, and drug abuse. As presented, therefore, the scope of the studies undertaken seems, from the outset, to be very large.

Functional, optical mapping, studies, as well as molecular and biochemical analyses of Nav channel subunit expression in human SAN, compared with human right atria (RA), are

presented. The approaches used are reasonable and potentially could be powerful, providing insights into the relationships between Nav channel expression patterns and functioning. The present manuscript falls short of doing this, however, as the numbers of samples/hearts analyzed, particularly in the optimal mapping section of the study/manuscript, are very small and no attempt is made to correlate the various data sets (i.e., the mapping, the molecular and the biochemical data). It is certainly appreciated that this sort of multiscale analyses is difficult. It also seems reasonable to assert, however, that to make a significant impact on the field, this would need to be done. In the absence of this sort of rigorous and robust experimental design, the experimental observations really are just that and they will remain as such until more robust experiments and analyses are completed.

As it stands, this manuscript presents two important findings, i.e., that Nav channels are functionally expressed in human SAN and, in addition that both cNav and nNav are expressed and functional. The molecular data presented suggest that the transcript expression levels of multiple Nav channel alpha and beta subunits are affected (and differentially so) by various conditions, from chronic alcohol consumption to heart failure. The functional consequences of these differential changes in transcript levels is not clear and, importantly, was not explored directly. The authors do present some molecular/computational modeling to begin to address the relationships between the molecular, cellular and circuit components and the role(s) of Nav channels in the SAN and RA. The findings presented are interesting, although interpreting the functional impact/significance is limited by the lack of experimental testing and validations of the model. In addition, the modeling efforts are limited in that the exclusive focus of the simulations presented was Nav channels; other ion channels, including ion channels likely to be differentially expressed in SAN and RA, appear not to be considered.

The authors might well be correct in their interpretations of the functional implications of their findings. There are, however, a number of concerns with the data presented and with the authors analyses and interpretations of the experimental data acquired that limit enthusiasm for this work as presented. Some of these concerns are enumerated below.

We thank Reviewer 2 for their positive comments and taking the time to provide constructive comments that helped us to significantly improve our manuscript and data presentation.

Major concerns:

1. As noted above, the initial studies presented are the optical mapping studies and the authors indicate that have mapped eight (n = 8) non-failing and two (n = 2) failing human hearts. On reading the text, it is unclear why mapping was done on only these 10 hearts when more were clearly available (and used for molecular and biochemical analyses). It appears, from the presentation of the optical mapping data that the data from these 10 hearts were combined and analyzed together, although no explanation or rationale is provided. Focusing on the main point of the paper, however, the authors do present data showing that perfusion of tetrodotoxin (TTX), a highly selective Nav channel blocker affects cycle length and conduction time, consistent with effects on conduction velocity, in isolated human SAN, at concentrations in the range of 100 nM to 3 μ M. Dramatic effects of TTX on conduction were also seen in preparations subjected to rapid atrial pacing or pre-treated with adenosine. It is unclear whether there are any detectable differences (in TTX sensitivity) between failing and non-failing hearts as, again, this point seems not to have been addressed (likely owing to the very small sample numbers). The rationale for the small sample numbers and for combining the functional (i.e., mapping) data acquired from non-failing and failing hearts remains unclear and unexplained.

Response: We apologize for any confusion in our manuscript presentation. As we previously mentioned in Limitations, “Few HF hearts were available for optical mapping studies due to bicaval surgical dissection which can impair SAN coronary artery perfusion.” Similarly, when lungs were used for transplantation, the posterior left atrial wall and interatrial septum would be removed from the donor heart, which can also damage/interrupt the SAN coronary artery. Although many intact SAN were available for molecular studies, only limited number of donor or failing hearts could be used for coronary-perfused SAN optical mapping.

Indeed, the small sample size of HF hearts does not allow meaningful statistical analysis to compare the functional differences of Nav in SAN pacemaking based on HF history. Thus, to show any Nav-related SAN dysfunction associated with disease, in the revised version, we have added new **Online Figure I** illustrating heart-specific responses to adenosine, atrial pacing (SNRT) with and without neuronal and cardiac Nav blockades. We also added this information to Results: “All functional data presented below are averages of all human SAN preparations studied with each specific drug protocol. Since the small sample size precludes subset analysis, heart specific data are presented in **Table 1** and **Online Figure I**”

2. It is unclear why the authors didn't design and complete experiments in which neuronal and

cardiac channels were selectively blocked, and the functional effects of these select manipulations were determined?

Response: The reviewer raises a very relevant question. At the time of the first functional experiment for this project, there were no validated blockers that could be used specifically for nNav 1.6 and nNav 1.1. Hence we used TTX at different doses previously shown to block these individual isoforms in animal studies¹. Fortunately, during the revision of this manuscript, we identified a specific blocker for nNav1.6, and conducted optical mapping experiments in four human SAN to test its individual contribution to SAN function. Data from these new experiments show that specific nNav1.6 channel blockade could depress SAN conduction and induce SAN exit block during adenosine bolus and overdrive suppression, which is comparable to TTX 100nM effects (**New Online Figure IV**). In one experiment we also tested selective nNav 1.1 channel blockade, but it did not show significant effects on SAN conduction or automaticity; hence, we did not continue to study the nNav1.1 blocker due to very limited number of intact human SAN available for functional experiments. In addition to our previous data set from molecular analyses, our new results show that nNav may contribute to human SAN pacemaking function, at least partially through nNav1.6 channel. We have added the description of these new data to Methods, Results and Discussion.

3. The authors do go on to examine the mRNA and protein expression levels of several Nav channel (alpha and beta) subunits using quantitative RT-PCR, immunostaining and western blot analyses of SAN tissue, compared with surrounding right atrial (RA) tissue (in the same hearts). It is unclear from the data presentation, however, how the molecular data relate to the functional data. It is similarly unclear how the biochemical data relate to the functional data, although this concern can be minimized by simply stating that the biochemical data should not be considered (i.e., interpreted) at all here, as only two Nav channel alpha subunits were examined.

Response: Thank you for this important question. As the reviewer suggests we agree that it would be important to understand the specific mechanisms by which Nav channels impact SAN automaticity and conduction. Since one of the main goals of this first-time study is to confirm the existence and function of both cNav and nNav in the human SAN, we focused on providing convincing data to quantify all known Nav isoforms and main subunits. For this study, the molecular and biochemical data are meant to complement the main functional findings from optical mapping studies by confirming the expression of these channels, and their alterations under different disease conditions.

However, as requested in your Comment #6, we quantified the expression levels of cNav1.5 and nNav1.6 in immunostained cryo-frozen tissue sections from three hearts functionally mapped with the nNav1.6 blocker (Hearts 957855, 283273, and 670263; new **Online Figure VII**). Our results show that in heart # 957855, which developed SAN exit block with nNav1.6 blocker, the ratio of nNav1.6 between SAN and RA was the highest compared to two other hearts, which exhibited less sensitivity to Nav1.6 blocker. These data may begin to explain the distinct functional contribution of nNav1.6 channels and correlate these findings with protein expression levels. These data are now included in Results and Discussion.

4. It is unclear why the authors restricted their molecular analyses to Nav alpha and beta subunits. Why, for example, weren't the expression levels of other Nav channel accessory/interacting protein subunits also examined? And, why weren't other channels (and receptors) that are likely differentially expressed in human SAN and RA also considered?

Response: Thank you for this important question. In the current study, we aim to “determine the existence and specific role of nNav and cNav isoforms in human SAN pacemaking and intranodal conduction...”, so we focused our molecular analyses on Nav alpha and beta subunits. We agree that there could be several Nav channel accessory/interacting protein subunits, which may also be important for SAN function, and prone to alterations in diseased hearts. While a large-scale investigation of the Nav channels and their accessory units may help provide a more comprehensive and mechanistic understanding of Nav channel function, these studies are out of the scope of the current study. However, in keeping with reviewer's request, we have now added new data to show quantitative transcript levels not only for Nav alpha and beta subunits but also HCN1 and HCN4, Cx40 and Cx43 as well as three calcium channels isoforms in the previously presented heart samples (n=20). These data are added to the **Online Table V** and Results.

Contrary to our interpretation of this suggestion, if the reviewer believes that analyses of accessory/interacting subunits are necessary and will help reveal novel aspects of Nav channel composition and function in this manuscript, we are willing to determine and include their transcript levels to our results.

5. One wonders why the authors did not use more comprehensive molecular strategies/ approaches to analyze the transcriptomes of their human SAN and RA tissues (i.e., rather than using RT-PCR to analyze one transcript at a time)?

Response: The reviewer raises a very valid query. We chose to use RT-PCR to detect and quantify the specific molecular targets of interest to test the central hypothesis for this study i.e. distribution of Nav channel isoforms in failing and non-failing SAN and RA. We agree that large-scale transcriptomic analyses may provide more detailed information about the overall transcriptomic profiles of the tissues studied, however, it may not provide quantification of less abundant protein transcripts including Nav isoforms, compared to those of other abundant proteins in the SAN e.g. HCN or ECM proteins. Hence we did not use other approaches to analyze transcriptomes, but rather focused on the channels/isoforms of interest using RT-PCR for this study. Moreover, adding large scale transcriptome sequencing data will overload this manuscript and make it unfocused. We are confident that our ongoing study utilizing next generation sequencing (NGS) techniques to study miRNA and mRNA profiles of human SAN and atria will generate large-scale comprehensive transcriptomic data that can shed light on several novel aspects of SAN function.

6. One similarly wonders why the authors did not attempt to correlate their individual data sets...i.e., the functional imaging and transcript expression data collected for the individual human hearts studied?

Response: Thank you for this valid question. It would certainly be ideal to be able to compare molecular data with functional studies and draw more convincing conclusions. So far, in order to ensure the best quality of RNA, we have preferred to use fresh frozen tissue from unmapped hearts for molecular studies, as several hours of *ex-vivo* optical mapping and multiple drug administration protocols may cause potential transcription level changes or RNA degradation. Furthermore, realistically there can be a further time-delay in dissecting these tissues after conclusion of the mapping experiments, since we have to remove the heart from the mapping setup and prepare for the elaborate and detailed dissection.

However, as per the reviewer's request, in order to correlate the functional findings with biochemical data, we conducted western blotting analyses for nNav1.6 on RA tissues collected

before functional experiments from ten human SAN preps used for optical mapping experiments. Unfortunately, most of the commercially available antibodies tested (Abcam #ab65166, Sigma#WH0006334M4 and Alomone#ASC-009) are unsuitable for human tissues and we could not detect the expected ~250kDa band. However, the Alomone antibody was reliable for immunostaining (as shown in **Figure 6**). Hence we quantified the expression levels of cNav1.5 and nNav1.6 in immunostained cryo-frozen tissue sections from three hearts functionally mapped with the nNav1.6 blocker (Hearts 957855, 283273, and 670263; Online Figure VII). Our results show that in heart # 957855, which developed exit-block with nNav1.6 blocker, the ratio of nNav1.6 between SAN and RA was the highest compared to two other hearts, which exhibited less sensitivity to nNav1.6 blocker. These data may begin to explain the distinct functional contribution of nNav1.6 channels and correlate these findings with protein expression levels. These data are now included in Results and Discussion.

7. It seems that the computer simulations of human SAN function and dysfunction are based on more assumptions, than actual data, and are, as a result, somewhat misleading.

Response: We agree with the reviewer that computer models, like all model systems, must make some assumptions. The cellular models used for SAN and atria were further adapted from the most recent and robust human SAN model² (Fabbri et al., *J Physiol* 2017) and the most widely used human atrial model³ (Countermahe et al., *Am J Physiol.*, 1998) computational studies, which were partially based on direct measurements of human myocardial ion channels. Beyond these models, we also made several specific adaptations. First, adaptations to the models, as described in **Online Table IV** were made to tailor the models to match the human optical mapping data baseline and during dose-dependent adenosine effect on human SAN pacemaking and conduction in the our current and recent studies⁴ (Li et al., *Science Translational Medicine* 2017). Second, the structure of the SAN model was based on the human-specific architecture of SAN conduction pathways detailed in⁵ (Csepe et al., *PBMS* 2017). Validation for these computer models comes from their ability to reproduce main functional observations. Importantly, with the validated computational human SAN models we were able to reveal the mechanism of the specific dysfunction observed which we could not elucidate in *ex-vivo* experiments. Specifically, we were able to reveal the contribution of selective Nav blockade in SAN-SACP vs atria in SAN dysfunction (Reviewer 1 comment) as well as the mechanism of absence entrance block during Nav blockade, when exit block

occurred (Reviewer 3 comment) due to direction-dependent safety factors of SAN conduction pathway.

Additional Specific Comments:

1. It is unclear what the authors mean (in the abstract) by noting that they have used “high-resolution” molecular mapping?

Response: We apologize for the confusing sentence. We have changed “using high-resolution optical and molecular mapping” to “using high-resolution optical mapping and molecular-biology techniques”.

2. The terms “conduction catastrophe” and “electrical stress” (also used in the abstract) are unlikely to be understood (or interpretable) by non-experts. The authors need to make the experiments, results and conclusions presented accessible to a general audience.

Response: We thank the reviewer for their suggestion, and changed those terms to “conduction failure” and “overdrive suppression”.

3. In presenting the analyses of their molecular (Nav channel subunit transcript expression) data), the authors repeatedly refer to “changes” in channel subunit expression levels. But the differences observed between different samples are not “changes”, they are simply “differences” as the same sample was not measured more than once, as would be needed to describe something as a “change”.

Response: Thank you for pointing this out. We have changed “changes” to “differences”.

4. The Discussion section of this manuscript is very long (6+ pages), particularly given the limited scope of the work, and could be reduced by 50% with no effect on the impact or import of the paper.

Response: According to the reviewer’s comment, we had reduced the overall discussion by 20% of words. But due to additional request for several new experiments and data, some sections of the discussion had to be elaborated/expanded. Overall we have tried to restrict the discussion to the main results of the manuscript.

Response to Reviewer #3:

This is a well-executed paper that helps to address an important scientific and clinical question, namely the role of the sodium current in cardiac pacemaking and why patients with sodium channel mutations may present with sinus node dysfunction.

We thank Reviewer 3 for their positive comments.

The term 'conduction catastrophe' is overly dramatic and lacks any explanation of the underlying physiology. I would encourage the use of a less emotive and more descriptive term.

Response: We have changed 'conduction catastrophe' to "conduction failure"

1. The sentence "Many patients with symptomatic SND are found to harbor loss-of-function mutations in the SCN5A gene" should read "some patients". SCN5A mutations are a recognised but rare cause of SND

Response: We have changed "Many patients" to "some patients".

2. Results: Heart rates of 56-116 bpm are stated to be in the normal range of intrinsic heart rates for humans this is not correct. Reference should be made to the paper "Jose & Collison, The normal range and determinants of the intrinsic heart rate in man, 1970" which should be considered the gold standard for intrinsic heart rates. Thus some of the hearts in this study have an intrinsic heart rate below the normal intrinsic rate. This does not detract from the scientific value of the paper.

Response: We thank the reviewer for this valuable suggestion. We have included the referenced paper, and removed the statement on normal heart rate.

3. Results: Control n=8 but TTX is n=7 / 6. Comment should be made as to why some of the control hearts did not go on to be included in the TTX dataset.

Response: In the first submission, a total of 10 hearts were used for optical mapping, but not all hearts were tested with both doses of TTX. We added to **Online Table I** to clarify the protocol used for individual hearts.

4. Results: "prolongation of SCL by cNav blockade was primarily due to depression of SAN conduction rather than automaticity." This does not make any sense in terms of the mechanisms of SAN automaticity. The authors should reference any data showing that SAN conduction velocity can alter automaticity or state that the mechanism by which this occurs is not clear.

Response: We are sorry for the confusion and agree with the reviewer. It is now stated as "prolongation of atrial CL by cNav blockade was primarily due to depression of SAN conduction rather than automaticity."

5. Results / methods: "Chronic Alcohol Consumption" – does this mean alcohol 'abuse' or drinking any alcohol at all. How was this data gathered for the clinical database? Was it from patients or relatives? Retrospective or prospective? How many units of alcohol, how long determines 'chronic' etc? More detail is needed for the methods

Response: "Chronic Alcohol Consumption" is defined as either abuse (>7 drinks/week for women and >14 drinks/week for men), as reported in documented medical records, or chronic consumption of moderate drinking (>2 drinks/week for at least 10 years). Alcohol consumption was first collected from medical records when available and then from retrospective interviews with family members by LifeLine of Ohio Organ Procurement Organization for rejected donor hearts. This information has been added to the Methods.

6. Results: Data is presented regarding SAN re-entry. This is interesting but the phenomenon of SAN re-entry is controversial. There is not sufficient data shown to allow the reader to make an interpretation of the maps to confirm re-entry is occurring. More detail should be given - eg videos showing the activation maps (including the atrial conduction), or at the very least detail of the interaction between the atrium and SAN during this phenomenon, perhaps it would be helpful if this data were 'zoomed in' on the area of re-entry to make a more convincing figure. I would also be interested to see the activation the SAN and the exit to the atrium from the SAN beat that is initiated by re-entry. Lastly in this regard, can the authors identify any critical features that lead to the conditions for re-entry? For example conduction time from exit to entry point? Conduction velocity in the atrium? Etc

Response: As requested by the reviewer, we have added additional extracted SAN optical action potentials to **Figure 4** as well as two **Online Movies I and II**, which show the post-pacing induced intranodal micro-reentry (**Movie I**) and SAN macro-reentry (**Movie II**), when the SAN activation exits to the atrium from the SAN from right inferior SACP and entrance back to the SAN through left inferior SACP. The mechanisms and conditions of SAN micro and macro-reentry were described in detail in our previous canine SAN optical mapping study Glukhov et al 2013 *Circulation Arrhythmia and Electrophysiology*⁶. We found that SAN reentry can occur during the first seconds of recovery from tachypacing during vagal or sympathetic stimulation in Glukhov et al 2013⁶, or during Nav blockade as in the current study.

Based on Glukhov et al⁶ and our current study, it is tempting to speculate that partial Nav blockade could heterogeneously affect the electrophysiologic features such as conduction velocity and refractory periods in SAN, SACP and surrounding RA. Specifically, preferential slowing of SAN conduction by Nav blockade, would lead to intranodal unidirectional blocks and initiate intranodal micro-reentry or macro-reentry. The micro-reentry pivot waves anchored to the longitudinal block region can produce both tachycardia and paradoxical bradycardia (due to exit block), despite an atrial activation pattern and ECG morphology identical to regular sinus rhythm. Intranodal longitudinal conduction blocks usually coincided with interstitial fibrosis strands and SAN artery as in the case of heart 118258 shown in **Figure 4 panel C** histology.

We found that in some cases atrial waves can directly initiate reentry by entering the SAN through one SACP, slowly rotating around the intranodal longitudinal block and then leaving the SAN through another SACP. High dispersion of refractoriness within the SAN enables desynchronized firing of neighboring pacemaker compartments. Thus, the earliest pacemaker activation wave can be blocked from propagating towards the neighboring compartment with longer refractoriness in the transverse direction but could propagate along the intranodal block. During the next 200 to 300 ms, recovery of the pacemaker compartment excitability in the compartment with longer refractoriness would allow the wave to cross the longitudinal block and initiate SAN reentry.

Depending on the excitable state of SACPs, 2 scenarios of reentry maintenance were observed. First (SAN macro-reentry), the reentrant wave could propagate through an active SACP and then activate the atrial myocardium. After 30-50 ms, the wave could re-entrant to SAN via another SACP and thus form a macro-reentry circuit with 2 main pathways: a slow path (6–8 mm) located between two SACPs and SAN intranodal pacemakers (~400–600 ms) and another

fast pathway (10–20 mm) located in the atria outside of the SAN between 2 SACP. Second (SAN micro-reentry), when conduction in SACP are depressed, SAN reentrant waves can circulate inside of the SAN around a longitudinal functional block (1–3 mm) usually coinciding with interstitial fibrosis strands and SAN artery. **Figure 4B and two Online Movies** show both scenarios.

In future, we plan to conduct simulation studies with human heart-specific SAN-atrial *in silico* models, which include anatomically accurate 3D SAN and atria to explore the conditions that lead to SAN reentry in a robust manner.

7. Discussion: It is interesting that TTX causes exit block but not entrance block. Can the authors speculate on why?

Response: Thank you for this intriguing comment. With the validated computational human SAN models we have been able to reveal the mechanism as to the absence of entrance block during slow atrial pacing in the presence of Nav partial blockade (20%) when exit block occurred. The mechanism is related to direction-dependent safety factors of SAN conduction pathway for exit vs entrance conduction. Computational analysis in control and HF human SAN models revealed that safety factor in SACP junction is direction-dependent and it is higher for entrance vs exit conduction at exit block conditions, which explains the absence of entrance blocks during slow atrial pacing and I_{Na} partial block (new **Online Figure XI**). Importantly, as we have previously shown⁴ the absence of entrance block in SACP during atrial pacing further inhibited excitability in the SACP and promoted occurrence of post-pacing exit block (**Online Figure XI panel B**) and SAN reentrant arrhythmias⁶, which can explain current experimental observations (**Figures 3B and 4**). We have added these new computational simulation data to **Online Figure XI** and Results page 13.

Reference List

1. Zimmer, T., Haufe, V., & Blechschmidt, S. Voltage-gated sodium channels in the mammalian heart. *Glob. Cardiol. Sci. Pract.* **2014**, 449-463 (2014).

2. Fabbri,A., Fantini,M., Wilders,R., & Severi,S. Computational analysis of the human sinus node action potential: model development and effects of mutations. *J. Physiol* **595**, 2365-2396 (2017).
3. Courtemanche,M., Ramirez,R.J., & Nattel,S. Ionic mechanisms underlying human atrial action potential properties: insights from a mathematical model. *Am. J. Physiol* **275**, H301-H321 (1998).
4. Li,N. *et al.* Redundant and diverse intranodal pacemakers and conduction pathways protect the human sinoatrial node from failure. *Sci. Transl. Med.* **9**, (2017).
5. Csepe,T.A. *et al.* Human sinoatrial node structure: 3D microanatomy of sinoatrial conduction pathways. *Prog. Biophys. Mol. Biol.* **120**, 164-178 (2016).
6. Glukhov,A.V. *et al.* Sinoatrial node reentry in a canine chronic left ventricular infarct model: role of intranodal fibrosis and heterogeneity of refractoriness. *Circ. Arrhythm. Electrophysiol.* **6**, 984-994 (2013).

REVIEWERS' COMMENTS:

Reviewer #1 (Remarks to the Author):

The authors have addressed my criticisms and questions in a satisfactory manner. The revised manuscript is significantly improved, especially by the new experimental data on expression levels of other proteins involved in pacemaking, and by the new simulations. I have no further comments.

Reviewer #2 (Remarks to the Author):

The authors are to be commended for the detailed and thoughtful response to the previous critiques. They have addressed the important issues identified and revised the manuscript accordingly. I still think that the Discussion section could be shorter, but this is a minor point. This paper makes an important contribution to the field.

Reviewer #3 (Remarks to the Author):

I am satisfied that my comments have been addressed.

REVIEWERS' COMMENTS:

Reviewer #1 (Remarks to the Author):

The authors have addressed my criticisms and questions in a satisfactory manner. The revised manuscript is significantly improved, especially by the new experimental data on expression levels of other proteins involved in pacemaking, and by the new simulations. I have no further comments.

Reviewer #2 (Remarks to the Author):

The authors are to be commended for the detailed and thoughtful response to the previous critiques. They have addressed the important issues identified and revised the manuscript accordingly. I still think that the Discussion section could be shorter, but this is a minor point. This paper makes an important contribution to the field.

Reviewer #3 (Remarks to the Author):

I am satisfied that my comments have been addressed.

RESPONSE TO REVIEWERS:

We thank all Reviewers for their positive endorsement of our manuscript and the effort they have invested to improve our study during the resubmission process. No changes were made except those which were requested by the Editor.